# Regulation of axonal morphogenesis by the mitochondrial protein Efhd1

Valeria Ulisse[1], Swagata Dey[1], Deborah E Rothbard[1] , Einav Zeevi[1], Irena Gokhman[1], Tali Dadosh[2], Adi Minis[1], Avraham Yaron[1]

During development, neurons adjust their energy balance to meet the high demands of robust axonal growth and branching. The mechanisms that regulate this tuning are largely unknown. Here, we show that sensory neurons lacking liver kinase B1 (Lkb1), a master regulator of energy homeostasis, exhibit impaired axonal growth and branching. Biochemical analysis of these neurons revealed reduction in axonal ATP levels, whereas transcriptome analysis uncovered down-regulation of *Efhd1* (EF-hand domain family member D1), a mitochondrial $Ca^{2+}$-binding protein. Genetic ablation of *Efhd1* in mice resulted in reduced axonal morphogenesis as well as enhanced neuronal death. Strikingly, this ablation causes mitochondrial dysfunction and a decrease in axonal ATP levels. Moreover, *Efhd1* KO sensory neurons display shortened mitochondria at the axonal growth cones, activation of the AMP-activated protein kinase (AMPK)–Ulk (Unc-51–like autophagy-activating kinase 1) pathway and an increase in autophagic flux. Overall, this work uncovers a new mitochondrial regulator that is required for axonal morphogenesis.

## Introduction

During development, neurons grow axons that elongate over substantial distances and branch for proper tissue innervation (Vaarmann et al, 2016). In humans, the axons of the peripheral nervous system (PNS) can reach lengths of up to 1 m (Misgeld & Schwarz, 2017). As axonal morphogenesis is energetically demanding, it must be supported by a tightly regulated energy balance.

Axonal ATP is produced primarily in the mitochondria, which are predominately localized in metabolically active zones of the neuron such as the growth cones at the leading edge of the axon (Vaarmann et al, 2016; Sheng, 2017). Mitochondrial function is critical to axonal morphogenesis; numerous reports have demonstrated that mitochondrial biogenesis, localization, trafficking, and local ATP production are all limiting factors for axonal growth and morphogenesis (Courchet et al, 2013; Spillane et al, 2013; Vaarmann et al, 2016; Misgeld & Schwarz, 2017). However, the regulatory mechanisms that couple axonal morphogenesis and energy supply remain poorly understood. The tumor-suppressor protein liver kinase B1 (Lkb1, also called Stk11) is a well-known regulator of cellular polarization in epithelia (Hardie, 2007; Shackelford & Shaw, 2009) and other non-neural tissues in *Drosophila* and vertebrates (Nakano & Takashima, 2012). In addition, studies in nonneuronal cells have established a critical function of the Lkb1 pathway in energy homeostasis mediated through enhancement of mitochondrial activity, mitochondrial biogenesis, and autophagy, as well as via a mammalian target of rapamycin-dependent decrease in energy expenditure and protein synthesis (Alexander & Walker, 2011; Hardie, 2011). Studies of the neuronal function of Lkb1 in the central nervous system (CNS) initially revealed its key role in establishing axon polarization and extension through the activation of the synapses of amphids defective kinases (Barnes et al, 2007; Shelly et al, 2007). More recently, deletion of *Lkb1* in the CNS revealed that it also contributes to axonal morphogenesis, in part through its effect on mitochondrial movement, biogenesis, and localization (Courchet et al, 2013; Spillane et al, 2013).

This study reports the discovery of a new pathway that couples energy homeostasis to axonal growth. In our investigation, we ablated the *Lkb1* gene in mice at the onset of PNS development. *Lkb1*-KO animals exhibited abnormal axonal growth and branching and reduced axonal ATP production. Intriguingly, transcriptome analysis of *Lkb1* KO sensory neurons uncovered significant down-regulation of the RNA transcript of the mitochondrial protein EF-hand domain family member D1 (Efhd1, also known as mitocalcin). Efhd1 is a calcium-binding protein that is localized to the inner mitochondrial membrane (Tominaga et al, 2006). To explore the function of Efhd1 in sensory neurons, we generated an *Efhd1* KO mouse line. Herein, we characterize these animals and demonstrate that Efhd1 regulates mitochondrial function and axonal morphogenesis during PNS development, providing a novel link of mitochondrial activity and energy homeostasis to axonal morphogenesis.

[1]Department of Biomolecular Sciences, The Weizmann Institute of Science, Rehovot, Israel  [2]Department of Chemical Research Support, Faculty of Chemistry, The Weizmann Institute of Science, Rehovot, Israel

Correspondence: Adi.Minis@rockefeller.edu; Avraham.Yaron@weizmann.ac.il
Adi Minis's present address is Strang Laboratory of Apoptosis and Cancer Biology, The Rockefeller University, New York, NY, USA

# Results

### Lkb1 KO sensory neurons display normal polarization but reduced axonal growth in vitro

To test the function of Lkb1 in the development of the PNS, we ablated the floxed *Lkb1* gene in the mouse at embryonic day 9 (E9) using the Wnt1–cre line, generating the strain henceforth referred to as *Lkb1* KO (Swisa et al, 2015) (Fig S1A). We first tested the polarization of dorsal root ganglion (DRG) neurons in vitro. After transfecting WT and *Lkb1* KO neurons with mCherry- and GFP-expressing plasmids, respectively, we cocultured the differentially labeled cells. This approach eliminates any effects that may arise from technical variations between the cultures or non-cell autonomous effects (such as secreted factors). Dissociated DRG neurons at E12.5 typically exhibit polarized morphology with a pair of axons growing from two opposite sides of the soma (Tymanskyj et al, 2018). Analysis of the *Lkb1* KO and WT neurons established that after 48 h, both cell types exhibit normal polarized morphology, with two axonal branches sprouting from opposite sides of the cell body (Fig S1B and C). These results support the conclusion of a previous study that suggested Lkb1 is dispensable for axon formation/polarization outside of the cortex (Lilley et al, 2013).

We next examined the importance of Lkb1 for axonal growth in vitro. DRG explants from E13.5 embryos were grown for 5 d embedded in 3D collagen matrix (Fig 1A and B). α-βIII-tubulin staining of the explants revealed that *Lkb1* KO axons were significantly shorter (50%) than those of the WT littermate controls (Fig 1E). Thus, unlike hippocampal and cortical neurons (Barnes et al, 2007; Shelly et al, 2007; Courchet et al, 2013), DRG sensory neurons are capable of establishing polarity in the absence of Lkb1, but their axon growth potential is significantly compromised.

### Lkb1 KO sensory neurons exhibit reduced axonal morphogenesis in vivo

Next, we assessed the role of Lkb1 in axonal morphogenesis in vivo. Limbs from E13.5 WT and *Lkb1* KO embryos were stained with α-βIII-tubulin and visualized by microscopy (Fig 1C and D). Compared with WT limbs, *Lkb1* KO limbs showed reduced axonal morphogenesis. Axonal morphology was quantified using NeuroMath (Rishal et al, 2013). Relative to those in WT mice, limbs in *Lkb1* KO mice displayed a strong reduction in the overall axonal coverage of the limb (36%) (Fig 1F) and an even more pronounced decrease in the total number of axonal branches (48%) (Fig 1G). To rule out the possibility that these observed phenotypes are secondary to neuronal death, we counted the number of Islet1 (a pan-sensory neuronal marker)-positive cells in DRGs of E15.5 embryos. We could not detect any significant difference in neuronal numbers between *Lkb1* KO and WT control littermates (Fig S1D), suggesting that the axonal abnormalities observed in the *Lkb1* KO mice were not caused by cell death. These results highlight the crucial role of Lkb1 as a regulator of developmental axonal morphogenesis in the PNS.

### Lkb1 KO neurons display reduced axonal ATP levels

As *Lkb1* ablation had no effect on sensory axon polarization, we hypothesized that the reduced axonal growth observed in the KOs in vitro and the reduced axonal morphogenesis phenotypes detected in vivo may be linked to the distinct function of LKB1 in metabolic homeostasis (Alexander & Walker, 2011; Germain et al, 2013; Hawley et al, 2003; Pooya et al, 2014). Because the *Lkb1* KO exhibits distinct axonal phenotypes, we determined the ATP concentration in different neuronal compartments using a filter culture system that allows us to differentially examine the cell bodies and the axons (Fig 1H). Whereas there was no difference observed between the ATP levels in *Lkb1* KO and WT soma, the ATP level was significantly reduced (49%) in the axons of *Lkb1* KO neurons compared with WT (Fig 1I). This dramatic decrease in ATP levels in *Lkb1* KO axons was not accompanied by activation of the metabolic sensor AMPK or its direct substrate acetyl CoA carboxylase (ACC), as judged by the extent of their phosphorylation (Fig S1E–G). This suggests that Lkb1 is a nonredundant activator of AMPK in sensory neurons. Taken together, these in vitro and in vivo results demonstrate that Lkb1 serves to maintain normal axonal development and ATP levels in mouse PNS neurons.

### Lkb1-deficient sensory axons display normal mitochondria motility

As mitochondria are the main source of axonal ATP and Lkb1 controls the axonal movement of the mitochondria in cortical neurons (Courchet et al, 2013), we tested the motility of mitochondria in sensory axons of DRG neurons from *Lkb1* KO and WT mice. *Lkb1* KO and WT neurons were plated on poly-D-lysine (PDL)/laminin–coated microfluidics chambers for 72 h (Maor-nof et al, 2016). The mitochondria were labeled by TMRE, which localizes to mitochondria, and the cells were imaged. No significant difference in mitochondrial motility was detected (Fig S1H–J).

### Lkb1 KO sensory neurons exhibit reduced expression of the mitochondrial protein Efhd1

In addition to directly phosphorylating its downstream targets, Lkb1 signals activation of gene transcription (Shackelford & Shaw, 2009). Therefore, to identify differential gene expression in DRGs directly isolated from E13.5 WT and *Lkb1* KO embryos, we performed transcriptome profiling using microarray analysis. Surprisingly, there were very few alterations in gene expression in *Lkb1* KO DRGs compared with WT controls. Most profound was the reduction in the expression of *Efhd1*, a mitochondrial $Ca^{2+}$-binding protein (approximately threefold) (Fig S2A and Table S1). In support, we detected a significant decrease in axonal Efhd1 protein expression in *Lkb1* KO neurons compared with WT (Fig 1J and L). To further test the connection between *Efhd1* expression and the Lkb1 pathway, we treated DRG neurons in vitro with the AMPK inhibitor compound C. After 8 h, we detected a profound reduction in axonal Efhd1 (Fig 1K and M). We have not detected a reduction in the levels of the mitochondrial protein TOM20 in the *Lkb1* KO or compound C–treated WT neurons (Fig S2B–E), suggesting that the reduction in Efhd1 is not due to global reduction in mitochondrial proteins.

Overall, these data show that Lkb1 pathway inhibition causes Efhd1 down-regulation in DRG neurons.

### Efhd1 is required for axonal growth in vitro

To investigate the physiological function of Efhd1, we generated an *Efhd1* KO mouse using the CRISPR–Cas9 technology (Fig S3A).

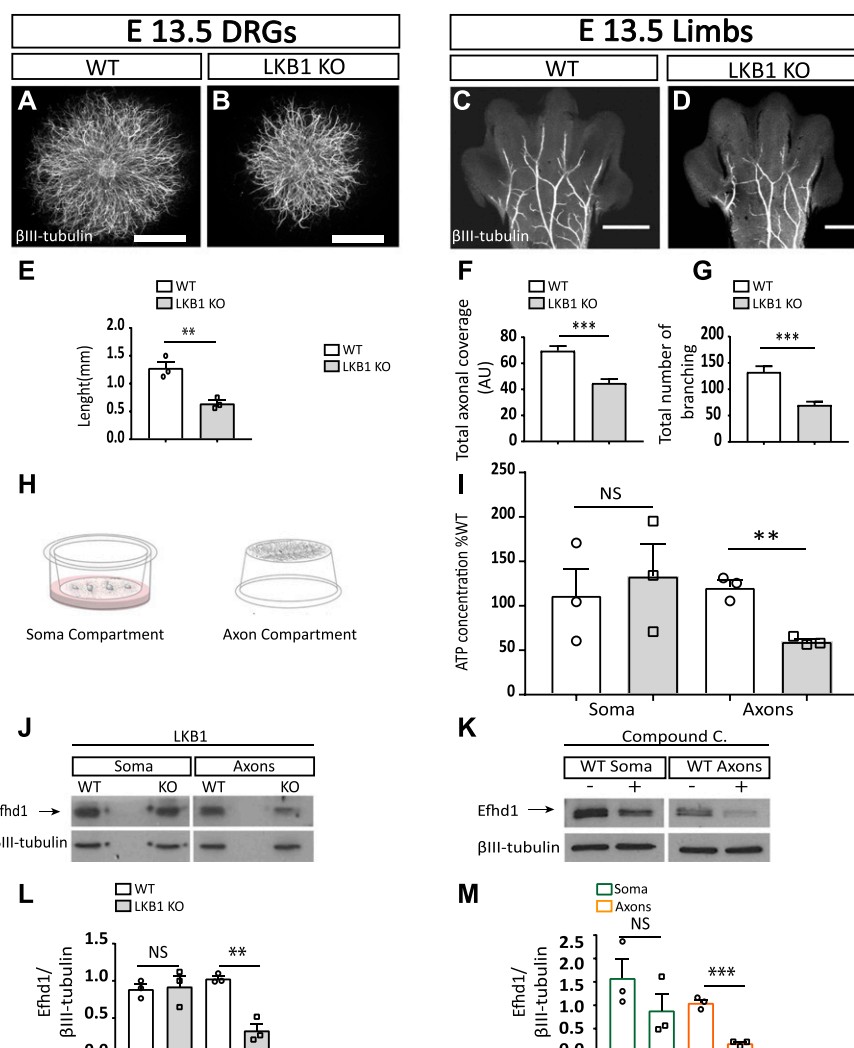

**Figure 1.   Liver kinase B1 (*Lkb1*) KO embryos display reduced axonal morphogenesis along with metabolic abnormalities and a decrease in the mitochondrial protein Efhd1.**

**(A, B)** Dorsal root ganglion (DRG) explants plated in 3D collagen. Scale bar: 1,000 *μ*m. **(C, D)** Limbs from WT and *Lkb1* KO E13.5 mouse embryos stained with *α*-*β*III-tubulin. Scale bar: 500 *μ*m. **(E)** Quantification of the axonal length of WT and *Lkb1* KO DRGs. Axonal lengths of eight WT and eight *Lkb1* KO DRG explants were quantified by four measurements in three independent experiments N = 3. Graphs show means ± SEM (unpaired *t* test \*\**P* = 0.0071). **(F)** Overall axonal coverage (total axonal length that cover the limb's surface). Graph shows means ± SEM (unpaired *t* test \*\*\**P* = 0.0001). **(G)** Total number of branching was quantified using NeuroMath. Graph shows means ± SEM (unpaired *t* test \*\*\**P* = 0.0001). **(F, G)** Eight WT (N = 8) and six *Lkb1* (N = 6) KO embryos were analyzed in (F, G); data for each embryo represent the average measurements of both limbs. **(H)** Schematic illustration of the filter cell culture system that facilitates the purification and biochemical analysis of cell bodies and axons. **(I)** Analysis of ATP levels in WT and *Lkb1* KO soma and axons. Graph shows means ± SEM of three independent experiments (N = 3) (unpaired *t* test: soma NS, unpaired *t* test: axons \*\**P* = 0.0019). Values are presented as the % of WT. **(J)** Immunoblot analysis of Efhd1 expressions levels in *Lkb1* KO and WT soma and axons. **(K)** Immunoblot analysis of Efhd1 expression in WT DRGs treated with 20 *μ*M compound C for 8 h. **(L)** Quantification of Efhd1 protein level in *Lkb1* KO DRGs compared to WT. Graph shows means ± SEM of three independent experiments (N = 3) (unpaired *t* test: soma NS, unpaired *t* test axons \*\**P* = 0.00232). **(M)** Quantification of Efhd1 expression in treated WT DRGs. Graphs show mean ± SEM of three independent experiments (N = 3) (unpaired *t* test: soma NS, unpaired *t* test axons \*\*\**P* = 0.0004). Source data are available for this figure.

Complete elimination of Efhd1 expression was confirmed by Western blot (Fig S3B). To determine if Efhd1 is required for axonal growth in DRG neurons in vitro, we first cultured DRG explants from WT and *Efhd1* KO E13.5 embryos for 48 h on PDL/laminin, which is highly permissive. Under these culture conditions, we have not observed any effect on axonal growth (Fig S4A–C). Next, we cultured E13.5 DRGs for 5 d in 3D collagen matrix. Axons were visualized by *α*-*β*III-tubulin staining (Fig 2A and B). Under these conditions, axons of the *Efhd1* KO neurons measured consistently shorter (18%) compared with those from WT littermates (Fig 2E). These results are reminiscent of the *Lkb1* KO phenotype and are consistent with previous studies on Efhd1 (Tominaga et al, 2006), supporting the notion that Efhd1 is required for axonal growth in vitro.

### *Efhd1* KO sensory neurons display aberrant axonal development and increased cell death in vivo

Next, we assessed whether Efhd1 is also required for axonal morphology in vivo. To this aim, we visualized *Efhd1* KO and littermate WT limbs of E13.5 and E14.5 embryos and quantified axonal morphology by NeuroMath. At E13.5, the axonal patterns in WT and KO appeared identical (Fig S4D, E, and H). However, at E14.5 (Fig 2C and D), we detected a significant 18% reduction in the overall axonal coverage (Fig 2F) and a 19% decrease in the total number of branches (Fig 2G). Similar results were obtained using *α*-neurofilament staining (Fig S4F, G, and I).

We then investigated whether the lack of *Efhd1* provoked neuronal loss. To assess the extent of cell death in our system, we stained for the pan-DRG neuronal marker, Islet1, and processed (i.e., active) caspase-3, which is an indicator for apoptotic cell death. No differences were noted at E15.5 between the numbers of neurons and the apoptotic rates in *Efhd1* WT and KO DRGs (Fig S4J and K). In contrast, at E17.5, a significant decrease in neuronal numbers (28%) was detected in *Efhd1* KO DRGs (Fig 2H–J), along with an increased rate of apoptotic cells (36%) (Fig 2L, M, and K).

Having demonstrated that the aberrant phenotypes in *Efhd1* KO mice begin to arise during early development, we examined

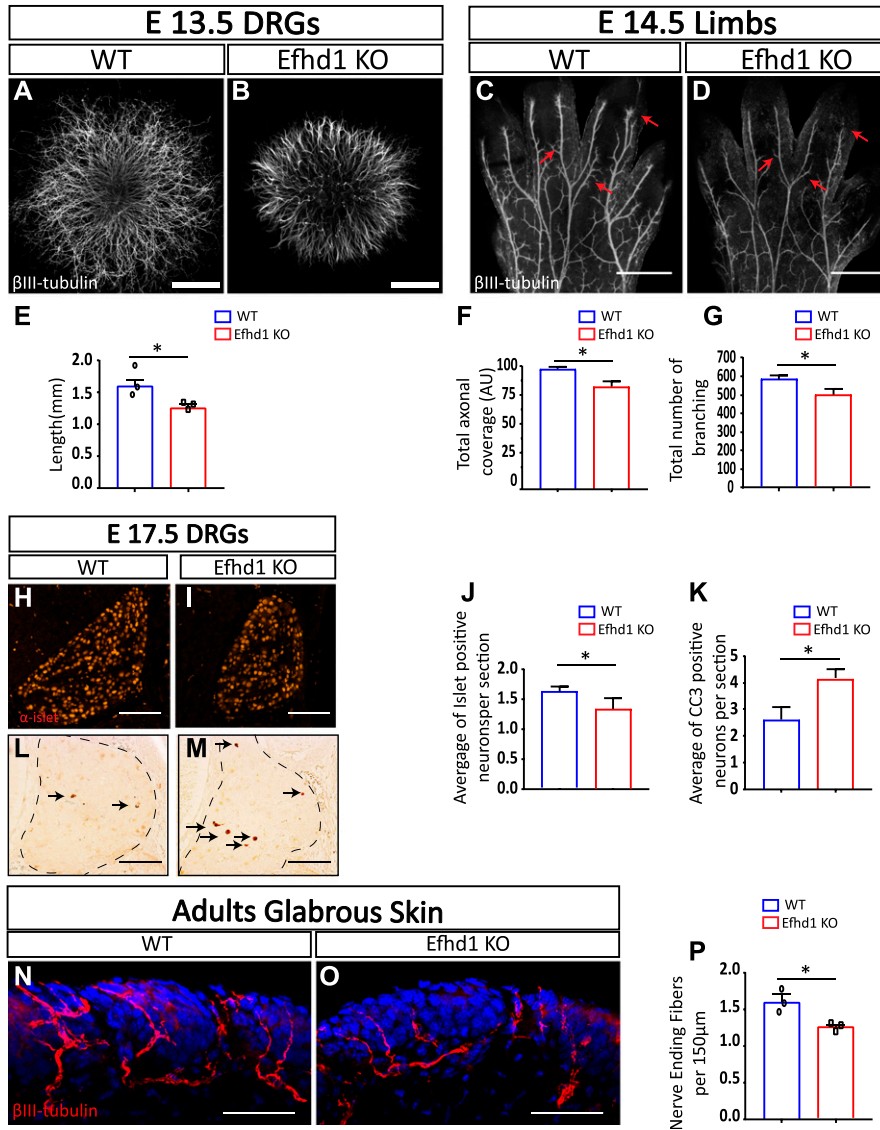

**Figure 2.  *Efhd1* KO neurons exhibit reduced axonal morphogenesis, neuronal cell loss, and decreased target innervation.**
**(A, B)** Dorsal root ganglion (DRG) explants placed in 3D collagen from WT and *Efhd1* KO E13.5 mouse embryos stained with α-βIII-tubulin. Scale bar: 1,000 μm. **(C, D)** Limbs from WT and *Efhd1* KO E14.5 mouse embryos, stained with α-βIII-tubulin. Arrows point missing axonal branches in the Efhd1 KO. Scale bar: 500 μm. **(E)** Axonal lengths of eight WT and eight *Efhd1* KO DRGs were quantified by four measurements in three independent experiments (N = 3). Graph shows means ± SEM (unpaired *t* test *P = 0.0237). **(F)** Overall axonal coverage (total axonal length that cover the limb's surface). Graph shows means ± SEM (unpaired *t* test *P = 0.038). **(G)** Total number of branching was quantified. Graphs show means ± SEM (unpaired *t* test *P = 0.032). **(F, G)** Seven WT (N = 7) and eight *Efhd1* KO (N = 8) embryos were analyzed in (F, G), data for each embryo represent the average measurements of both limbs. **(H, I)** E17.5 WT and Efdh1 KO mouse embryos were stained with anti-Islet1. Scale bar: 100 μm. **(J)** Graph shows means ± SEM (unpaired *t* test *P = 0.011). **(L, M)** E17.5 WT and Efdh1 KO mouse embryos were stained with anti-cleaved caspase-3 (CC3). Scale bar: 100 μm. **(K)** Graph shows means ± SEM (unpaired *t* test *P = 0.020). Arrows point to the CC3-positive cells in the DRGs. **(J, K)** Five WT (N = 5) and seven *Efhd1* KO (N = 7) embryos were analyzed, and the numbers of Islet1-positive neurons (J) and of CC3 positive (K) were quantified in 60 sections/embryo. **(N, O)** Glabrous skin of adult hind limbs from WT and Efdh1 KO adult mice was stained with α-βIII-tubulin and visualized by confocal microscopy. Scale bar 100 μm. **(P)** Nerve-ending fiber quantification. Graph shows means ± SEM (unpaired *t* test *P = 0.013). 25 sections from each left hind limb of three WT (N = 3) and three *Efhd1* KO (N = 3) adult animals were analyzed.

whether innervation in adult animals is also affected. After staining the hind limb glabrous skin with α-βIII-tubulin antibody (Fig 2N and O), we detected a significant reduction (33%) in the number of terminal fibers in *Efhd1* KO hind limb skin compared with the WT control (Fig 2P). Overall, these data suggest that Efhd1 is required in sensory neurons for axonal morphogenesis during development and for proper target innervation.

### *Efhd1* KO axons manifest lower ATP levels and shortened mitochondria

As Efhd1 is a mitochondrial protein, we examined whether its ablation affected energy homeostasis of the neuron. We measured ATP levels in WT and *Efhd1* KO neurons from E13.5 embryos and found that whereas no significant differences in ATP levels were observed

between the corresponding WT and KO DRGs somas, the ATP levels of Efdh1 KO axons are much lower (49%) than that of WT axons (Fig 3A).

Having established that axonal ATP is impacted by *Efhd1* deficiency, we reasoned that mitochondria numbers or morphology might also be affected. We examined mitochondria number in *Efhd1* KOs using two approaches. First, we directly counted the number of mitochondria at the axonal growth cones of DRG explant (see schematic illustration in Fig 3B) after visualizing them using super-resolution microscopy. No significant difference in mitochondrial number was observed between the WT and Efnd1 KO DRGs (Fig 3C). Next, we quantified the total mitochondria mass in DRG neurons by conducting real-time PCR analysis of six mitochondrial genes (Maryanovich et al, 2015; Ruggiero et al, 2017). No significant differences were detected between WT and Efnd1 KO DRGs by this approach as well (Fig 3D).

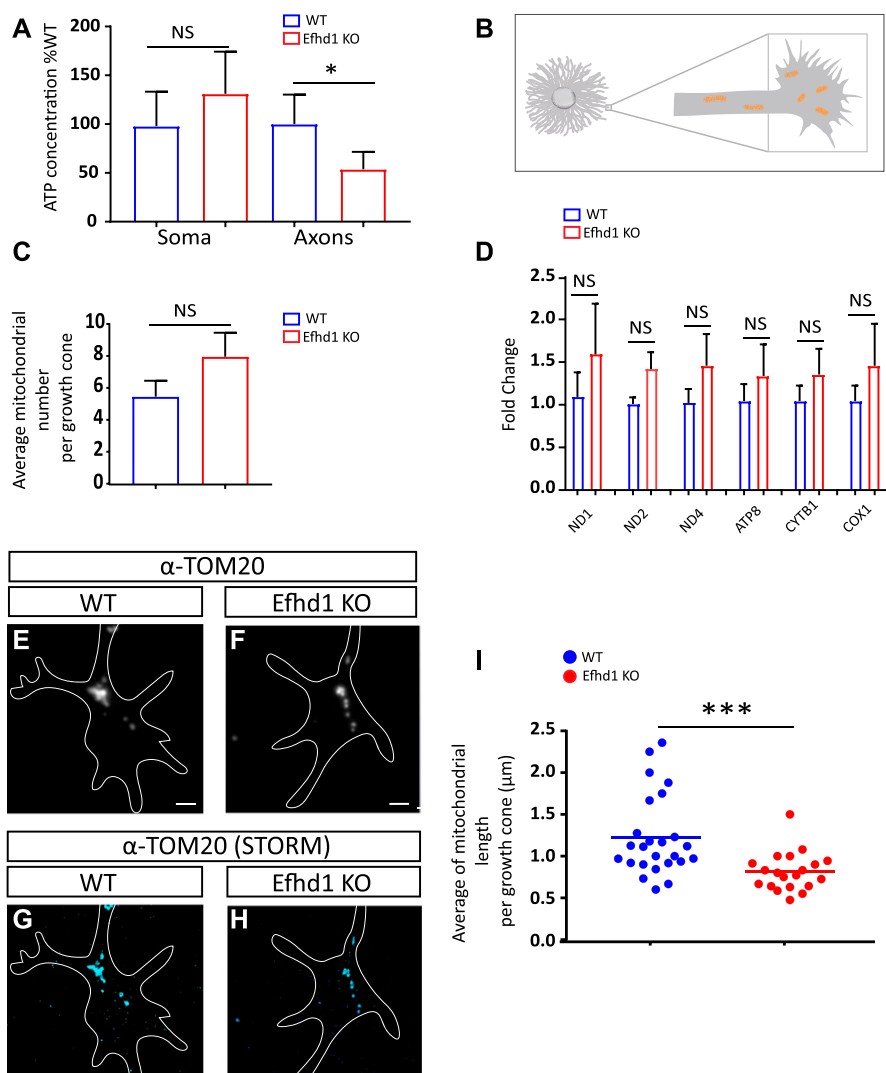

**Figure 3. *Efhd1* KO axons exhibit decreased ATP levels and shortened mitochondria.**
**(A)** Soma and axonal ATP levels of cultured dorsal root ganglion neurons were measured using the filter cell culture system. Values are presented as the % of WT. Graph shows means ± SEM of 12 independent experiments (N = 12) (Wilcoxon signed rank test: soma = NS, axons *P = 0.0034). **(B)** Schematic illustration of the plated dorsal root ganglion explant, axon and growth cone (marked by square) from which mitochondria morphology and number were analyzed by TOM20 staining. **(C)** Quantification of the number of mitochondria per growth cones counted in the super-resolution images. Graphs show means ± SEM of 25 growth cones WT and 20 *Efhd1* KO (unpaired *t* test NS). **(D)** Real-time PCR analysis of six mitochondrial genes (ND1, NADH dehydrogenase subunit 1; ND2, NADH dehydrogenase subunit 2; ND4, NADH dehydrogenase subunit 4; ATP8, ATP synthase protein 8; CYTB1, cytochrome b COX1: cytochrome c oxidase subunit 1). Graph shows means ± SEM of four independent experiments (N = 4) (unpaired *t* test all NS). **(E, F)** Wide-field fluorescence image of anti-TOM20 staining. Scale bar 2 *μ*m. **(E, F, G, H)** Super-resolution (STORM) images of anti-TOM20 staining of the same area of (E, F). Scale bar 2 *μ*m. **(I)** Quantification of mitochondria length: each point represents the average mitochondria length in a single growth cone; >150 mitochondria were analyzed from each genotype, N = 25 WT growth cones and N = 20 Efhd1 KO growth cones. Graph shows means ± SEM (Mann–Whitney test: ***P = 0.0002).

In parallel, we examined mitochondria morphology of the above samples. The mitochondria in *Efhd1* KO growth cones were significantly shortened compared with those in WT organelles (0.77 versus 1.13 *μ*m, respectively) (Fig 3E–I). Together, these results demonstrate that lack of *Efhd1* results in metabolic dysfunctions in sensory axons, which correlates with reduced axonal morphogenesis and aberrant mitochondria morphology.

### Ablation of *Efhd1* causes mitochondrial dysfunction in sensory neurons

Our observations that *Efhd1* KO neurons have similar number of axonal mitochondria with irregular morphology prompted us to directly examine the mitochondrial oxidative phosphorylation (OXPHOS)-mediated ATP production activity in these cells. We used the oxygen consumption rate, as measured by the Seahorse XF96 system, as a readout of OXPHOS. An equal number of E13.5-dissociated DRGs were plated on PDL/laminin for 4 d, and the mitochondrial activity measurements were preformed according to

Styr et al (2019). The *Efhd1* KO neurons exhibited clear reduction in the basal mitochondria respiration (Fig 4A–C), ATP-linked respiration, and ATP production (Fig 4A and D). We also detected significant reduction in the spare respiratory capacity (SRC) based on multi-comparison Sidak's test (Fig 4B). However, when the SRC was calculated according to the baseline of each genotype, we did not detected any significant difference (Fig 4E). Therefore, we cannot conclude that the SRC is defective in the *Efhd1* KO. No differences were detected in the non-mitochondrial respiration (proton leak) activity (Fig 4A, B, and F). Overall, our data show that Efhd1 is required for mitochondrial activity and ATP production under basal conditions in sensory neurons.

Last, we tested if the deficits in mitochondrial activity are also associated with changes in mitochondrial motility. *Efhd1* KO and WT neurons were plated on PDL/laminin-coated microfluidics chambers for 72 h (Maor-nof et al, 2016). The mitochondria were labeled by MitoTracker, which localizes to mitochondria regardless of their membrane potential, and the cells were imaged for 6 h (Ionescu et al, 2016). We did not detected any change in the

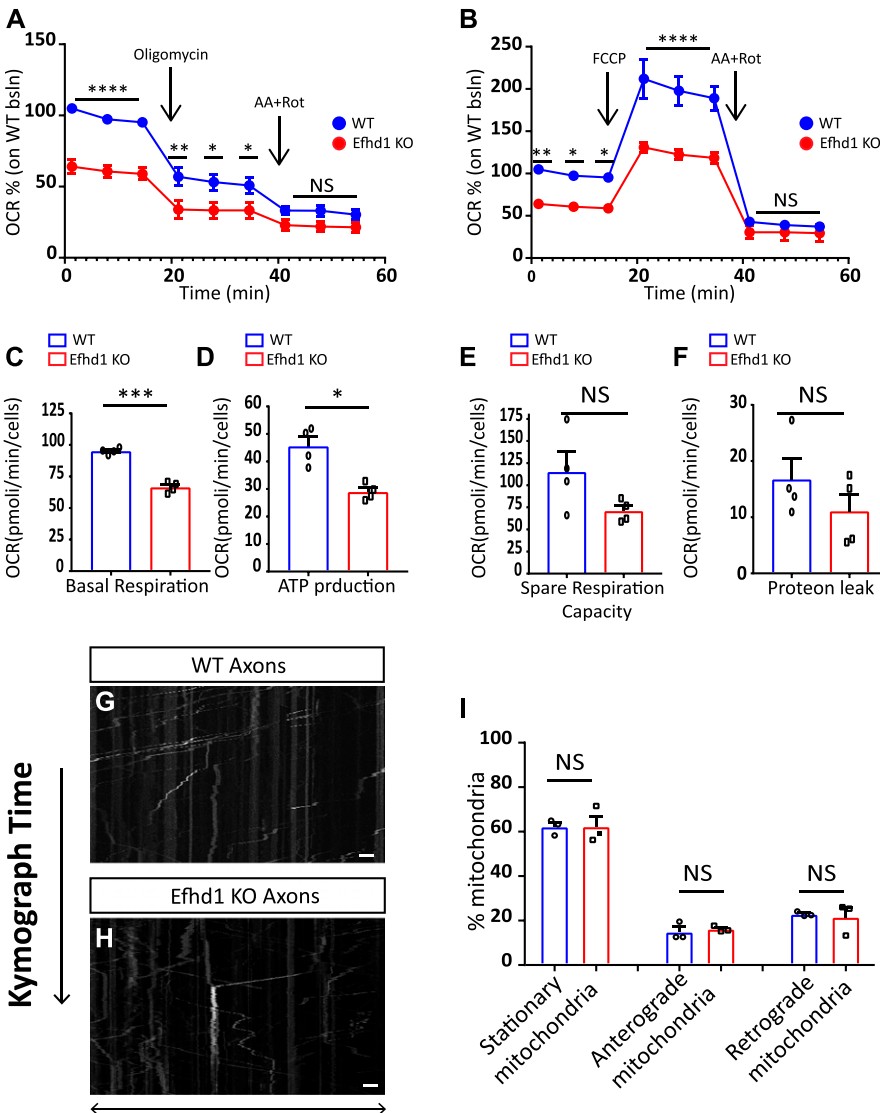

**Figure 4. *Efhd1* KO neurons show decrease in mitochondrial activity.**
**(A, B)** Analysis of the oxygen consumption rates (OCRs) in dissociated dorsal root ganglion (DRG) from E13.5 WT and *Efhd1* KO embryos by the Seahorse Bioscience XF96 analyzer. **(A, B)** First, the basal respiration was measured, then (A) oligomycin (1 $\mu M$) was added to measure the ATP production, and (B) FCCP (4 $\mu M$) was added to measure the spare respiratory capacity. The respiration was stopped and non-mitochondrial oxygen consumption was measured after injection of 0.5 $\mu M$ rotenone (Rot) and 0.5 $\mu M$ antimycin A (AA). Data are presented as the % of the WT baseline (bsln). The analysis is based on four independent biological experiments, (N = 4) in each experiment, at least nine wells were used for each condition for each genotype. **(A, B)** Two-way ANOVA test was performed with Sidak's multi-comparison test: (A) Baseline point 1-2-3,****$P \leq 0.0001$, Oligomycin point 1 **$P = 0.004$, point 2 *$P = 0.0195$ point 3 *$P = 0.0462$, rotenone and antimycin $P =$ NS; (B) baseline point 1 **$P = 0.0041$, point 2 *$P = 0.0132$, point 3 *$P = 0.0147$, FCCP point 1-2-3 ****$P \leq 0.0001$, rotenone and antimycin $P =$ NS. **(C, D, E, F)** Graphical quantification of (C) basal respiration, (D) ATP production, (E) spare respiratory capacity, and (F) non-mitochondrial ATP production (proton leak) (C: unpaired *t* test ***$P = 0.001$, D: unpaired *t* test **$P = 0.0042$, E, F: unpaired *t* test NS). Graphs show means ± SEM of OCR that are based on four independent biological experiments (N = 4), in each experiment, at least nine wells were used for each condition for each genotype. **(G, H)** Mitochondrial kymograph of axonal mitochondria motility in dissociated neurons from E13.5 DRGs WT and *Efhd1* KO. Scale bar 2 $\mu m$. **(I)** Quantification of the percentage of stationary, anterograde, and retrograde mitochondria in WT and *Efhd1* KO neurons. All graphs show mean ± SEM based on three experiments (N = 3); DRGs were dissected from three WT (N = 3) and three *Efhd1* KO (N = 3) embryos (stationary mitochondria: unpaired test NS, anterograde mitochondria: unpaired *t* test NS, retrograde mitochondria: unpaired *t* test with Welch's correction NS).

overall pattern of mitochondrial motility in the *Efhd1* KO neurons (Fig 4G–I).

### *Efhd1* KO sensory neurons have activated metabolic stress signaling

The mitochondrial dysfunction observed in Efdh1 KO axons prompted us to examine the activation status of AMPK in those neurons. We discovered that, unlike in *Lkb1* KOs, AMPK is hyper-phosphorylated in *Efhd1* KO axons and DRGs as compared with the WT control, both in vitro (Figs 5A and C and S5A) and in vivo (Figs 5B and F and S5B). Consistently, we also detected higher phosphorylation of the canonical AMPK target, ACC, in the soma and axons of *Efhd1* KO DRGs compared with WT (Figs 5A and D and S5C). As AMPK also regulates the recycling of aberrant mitochondria via mitophagy through stimulation of Ulk (Toyama et al, 2016; Zhang & Lin, 2016), we tested whether this pathway is

enhanced in *Efhd1* KOs. In line with AMPK activation, Ulk serine-555 phosphorylation was markedly elevated in the *Efhd1* KO DRGs both in vitro (Figs 5A and E and S5D) and in vivo (Figs 5B and G and S5E).

Next, we tested if there is an increase in autophagic flux in *Efhd1* KO neurons. We cultured WT and *Efhd1* KO DRG neurons in the presence of the lysosomal inhibitor bafilomycin. Under these conditions, autophagic vesicles are not degraded, allowing accumulation of the lipidated form of the LC3 protein (Zhang et al, 2012; Jiang & Mizushima, 2015; Redmann et al, 2017). Although, as expected, bafilomycin caused an increase in the levels of lipidated LC3 (LC3-II) in WT neurons, we detected a further increase in LC3-II levels in the *Efhd1* KO neurons (Fig 5H and I). These results support the notion that autophagic flux is enhanced in *Efhd1* KO cells, presumably to recruit and recycle the aberrant, dysfunctional mitochondria. This agrees with our finding that AMPK and Ulk are activated in *Efhd1* KO cells. Overall, these results suggest that *Efhd1*

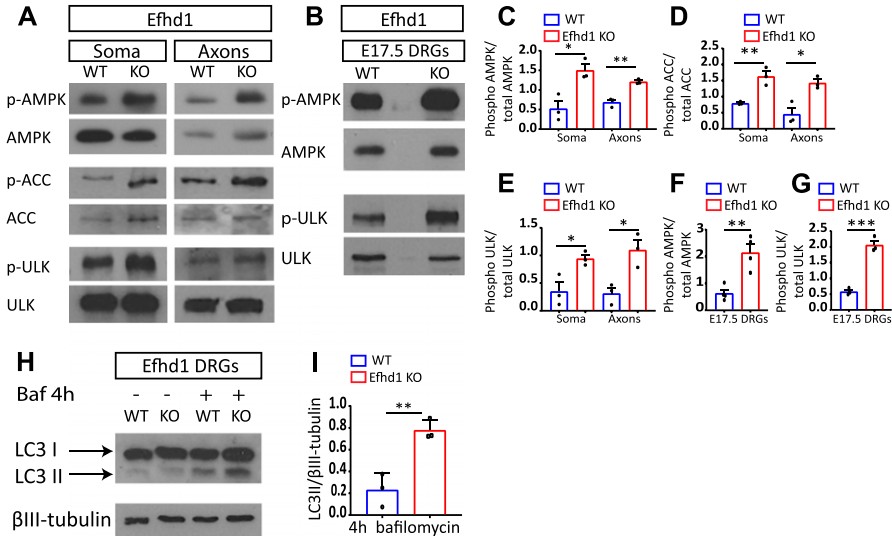

**Figure 5. *Efhd1* KO sensory neurons display AMPK and Ulk activation along with increased autophagic flux.**
**(A)** Immunoblot analysis of AMPK and Ulk activation in E13.5. Dorsal root ganglion (DRG) explants were grown on the filter cell culture system for 48 h. AMPK (Thr172), ACC (Ser79), and Ulk (Ser555) phosphorylation status were analyzed. **(B)** AMPK (Thr172) and Ulk (Ser55) phosphorylation status were analyzed by immunoblot of in vivo (directly extracted) E17.5 DRGs. **(C, D, E, F, G)** The ratio of phosphorylated to total levels of protein normalized to tubulin was quantified using ImageJ levels in E13.5 (C, D, E) and E17.5 (F, G) DRGs was quantified using ImageJ. **(C, D, E)** AMPK: soma *P = 0.0168, axons **P = 0.0022, (D) ACC: soma **P = 0.0069, axons *P = 0.0131, (E) Ulk: *t* test soma *P = 0.0287, axons *P = 0.0185. Graphs show means ± SEM based on three independent experiments (N = 3), unpaired *t* test was used for all. **(F, G)** AMPK: *t* test **P = 0.0038 (G) *t* test ***P = 0.0005. All graphs show means ± SEM based on N = 4 (p-AMPK) and N = 3 (p-Ulk) independent experiments, unpaired *t* test was used for both. **(H)** Immunoblot of LC3I and LC3II levels of WT and *Efhd1* KO DRGs treated with bafilomycin (100 nM) for 4 h. **(I)** The ratio of LC3-II to tubulin was quantified with ImageJ. Graphs show means ± SEM based on three independent experiments (N = 3) (unpaired *t* test **P = 0.006).
Source data are available for this figure.

ablation triggers the activation of metabolic stress pathways in an attempt to compensate for metabolic deficits.

## Discussion

In this study, we have identified a metabolic regulator of axonal morphogenesis. We generated and characterized a mouse model that lacks the key metabolic regulator Lkb1. Our data demonstrate that when *Lkb1* is ablated in the PNS during early development, neurons display reduced axonal growth both in vivo and in vitro. Cultured *Lkb1* KO neurons exhibit a decrease in axonal ATP levels (Fig 6A). Despite this decrease and the consequently elevated cellular stress level, we did not detect increased phosphorylation of AMPK, which is considered to be the sensor of cellular stress (Hawley et al, 2003; Hardie, 2007). These data are in line with the

abundance of reports demonstrating that Lkb1 is the principal AMPK kinase (Hawley et al, 2003; Shackelford & Shaw, 2009; Alexander & Walker, 2011).

To identify new regulators of energy homeostasis, we searched for alterations of gene expression in *Lkb1* KO cells. The expression of Efhd1 was significantly reduced in the KO neurons. We also found it to be down-regulated in sensory neurons upon pharmacological inhibition of AMPK. These results imply that Efhd1 might be an effector of the Lkb1–AMPK pathway. Additional studies are required to establish if the Lkb1–AMPK pathway directly controls Efhd1 levels, and if so, the exact mode of regulation.

Our results suggest that Efhd1 is required in sensory neurons for mitochondrial morphology and function. Interestingly, effects of its ablation are mostly manifested in the axons. This may be due to the fact that axons are more sensitive to mitochondrial dysfunction because of their length or because other metabolic pathways

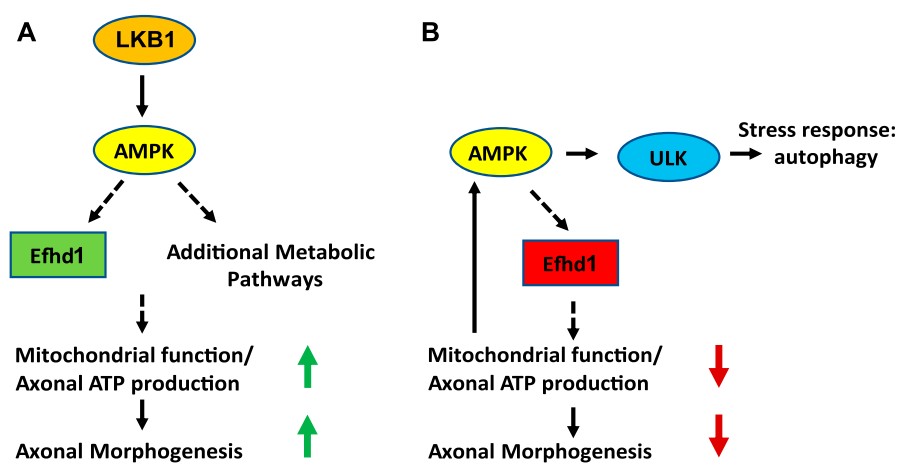

**Figure 6. Regulation of axonal morphogenesis by Efhd1.**
**(A)** Lkb1 controls mitochondrial homeostasis and axonal ATP production, through Efhd1 and additional pathways that are required for axonal morphogenesis. **(B)** Complete ablation of *Efhd1* causes reduction in the axonal ATP levels and mitochondrial abnormalities. This results in the activation of AMPK and Ulk and increased autophagic flux, which is correlated with reduced axonal morphogenesis.

compensate more efficiently in the soma. Our in vivo analysis also supports the idea that axons are more sensitive to the lack of *Efhd1*, as we detected axonal phenotypic changes as early as E14.5, whereas enhanced cell death was only observed at E17.5.

Our biochemical analysis shows that the *Efhd1* KO neurons respond to mitochondrial dysfunction by activating metabolic stress pathways, as manifested by the hyper-phosphorylation of AMPK. Although activated AMPK normally rescues the metabolic cellular function in stress condition (Gwinn et al, 2008; Toyama et al, 2016), in late stages of DRGs development (E17.5), we observed an increased incidence of apoptotic death of *Efhd1* KO cells, suggesting that the energy imbalance at this stage cannot be completely resolved despite enhanced AMPK activity. Consistent with the higher apoptotic rate observed at E17.5, innervation of the adults *Efhd1* KO skin was significantly sparser compared with WT.

The presence of aberrantly dysfunctional mitochondria in the *Efhd1* KO axons raises the notion that mitophagy might be activated in *Efhd1* KO DRGs. Indeed, we found enhanced Ulk phosphorylation at serine-555, which facilitates mitophagy in response to AMPK activation and increases the overall autophagic flux (Toyama et al, 2016) (Fig 6B). In line with our results, a recent study revealed that mitochondria shortening and the fission process can be directed by AMPK through the mitochondrial fission factor protein. Once AMPK is activated in response to cellular stress, the mitochondrial fission factor is phosphorylated and promotes the constriction and fission of mitochondria, prompting their engulfment by mitophagosomes (Toyama et al, 2016; Zhang & Lin, 2016).

The exact mitochondrial function of Efhd1 remains to be discovered. Previous studies linked Efhd1 to metabolic regulation in the development of bone marrow cells (Stein et al, 2017) and to the activation of mitoflashes, short stochastic superoxide bursts during cellular respiration (Hou et al, 2016). Notably, Efhd1 modulation was described earlier in a neuronal cell line where Efhd1 overexpression resulted in neurite extension, whereas its down-regulation caused neurite reduction and subsequent cell death (Tominaga et al, 2006). Given its calcium-binding capacity (Tominaga et al, 2006), it is tempting to speculate that Efhd1 may serve as a new link between the mitochondrial $Ca^{2+}$ and the mitochondria OXPHOS. Multiple studies have clearly demonstrated that $Ca^{2+}$ is a physiological regulator of OXPHOS and that it can facilitate the activation of several enzymes in the tricarboxylic acid (TCA) cycle (Rizzuto et al, 2012). Upon $Ca^{2+}$ binding by its EF-hand domain, Efhd1 may directly interact with proteins of the OXPHOS machinery in the mitochondrial inner membrane and stimulate their activity. Then, when Efhd1 function is impaired, either partially or completely (as in *Lkb1* KO and *Efhd1* KO, respectively), this $Ca^{2+}$-dependent regulation of ATP production would be compromised. This may trigger mitochondrial stress due to unmet energy demand during axonal growth, provoking mitochondria recycling through mitophagy.

Despite their sensory neurons' mitochondrial dysfunction, resultant aberrant axonal growth, and neuronal loss, mice with *Efhd1* deficiency are viable, are fertile, and have normal appearance and behavior in the cage. Moreover, *Efhd1* KO DRGs and axons display only limited defects in vitro and in vivo, highlighting the specific contribution of Efhd1 to axon-remodeling processes but not to basal homeostasis of the soma.

This study presents a new pathway of mitochondrial regulation. Lkb1, already known as a master metabolic regulator (Shaw et al, 2004; Shackelford & Shaw, 2009; Alexander & Walker, 2011; Pooya et al, 2014) modulates expression of the mitochondrial protein Efhd1. As demonstrated herein, Efhd1 has a unique and crucial role in axonal development of the PNS.

# Materials and Methods

### Mouse strains: Lkb1 and Efhd1

The *Lkb1* conditional KO mouse line was established by crossing *Lkb1* flox mice (Swisa et al, 2015) with Wnt1–Cre transgenic mice. The Wnt1–Cre transgene is expressed in neural crest cells (progenitors of [DRG] neurons and glia), and in the midbrain and dorsal neural tube of the CNS (Charron et al, 2003). Wnt1 expression starts at the embryonic day 8 (E8) in the midbrain and is expressed in its full pattern by E9 (Charron et al, 2003). The *Lkb1* KO mice die shortly after birth.

The *Efhd1* KO mouse line was generated by CRISPR–cas9 technique. The following two guide RNAs were used to delete a part of the *Efhd1* gene coding region and its promoter:

> Guide 1: sgRNA1-top: CACCgGAGGTTCGCGATCCGGTACG, sgRNA1-bottom: AAACCGTACCGGATCGCGAACCTCc.
> Guide 2: sgRNA2-top: CACCgGTTCAGTTCGGAGTCCGCGC, sgRNA2-bottom: AAACGCGCGGACTCCGAACTGAACc.

The *Efhd1* KO mice are viable and appear indistinguishable from their WT sibs in cage environment. Mice are of mix genetic background in all of the experiments, and we used littermate (WT) controls. Mice were hosed in specific pathogen-free facility, under 50% humidity, 12-h light/dark cycle, $22^c$ with standard diet (18% protein, 5% fat). All animal experiments followed protocols approved by The Weizmann Institute of Science Institutional Animal Care and Use Committee.

### Explant culture, medium, and culture method

DRG explants of E13.5 mice were aseptically removed from E13.5 embryos in L15 medium (L15 powder [MFCD00217482; Sigma-Aldrich]) dissolved in filtered distilled water (FDW) with 5% fetal bovine serum (10091148; Gibco). Chambers and filter plates: insert (cell culture insert, six-well hanging insert, 1 μm PET, MCRP06H48; Millicell), eight chambers (cell culture slide eight well, 30108; Life Science Co. Ltd.), six-well plate (140675; Thermo Fisher Scientific), and cell culture dish (430156; Merck). The coating was done for 1 h with PDL (P6407; Sigma-Aldrich) diluted in FDW, final concentration 0.01 mg/ml and, after washing briefly with FDW, laminin (114956-81-9; Sigma-Aldrich), and diluted in filtered F12 (01-095-1A; Biological Industries) for 2 h at 37°C at a final concentration of 10 μg/ml. The plated explants were grown for 48 h in complete NB (10888022; Thermo Fisher Scientific) supplemented with 2% B27 (17054-044; Thermo Fisher Scientific), 1% penicillin–streptomycin solution (03–0311B; Biological Industries), 1% glutamine (25030081; Gibco), and 12.5 ng/ml NGF (CAS 866405-64-3).

For protein analysis, DRGs were plated in the insert, growth for 48 h in NB plus NGF, and the cell bodies and the axons were lysed

followed by Western blot of the samples. When indicated, 20 $\mu M$ compound C (CAS 866405-64-3; Sigma-Aldrich) for 8 h or 100 nM bafilomycin (LC Laboratories) for 4 h was added after 48 h of cell growth in NB. The cultures were then lysed and processed for Western blot.

For immunofluorescent staining, DRGs were plated in eight chambers or cell culture plates for 48 h in NB plus NGF, fixed with 4% formaldehyde for 1 h at room temperature, and stained using mouse tubulin $\beta III$ antibody. Images were taken with DS-Qi2 fluorescent microscope, Nikon.

### 3D collagen cultures

Collagen cultures were performed as previously described (Charron et al, 2003; Romi et al, 2014). Briefly, DRGs, dissected as described above, were embedded in 2 mg/ml collagen matrix (1179179001; Roche) supplemented with NB plus NGF. After 5 d in culture, the explants were fixed with 4% formaldehyde and stained using mouse tubulin $\beta III$ antibody. DRGs were visualized with Leica MZ16F binoculars (Nikon), axonal length was measured using ImageJ software.

### Sample lysis and immunoblot quantification

Two different cell lysis buffers were used for Western blot analysis:

1) RIPA buffer (50 mM Tris, pH 7.4, 150 mM NaCl, 1% NP40, 0.1% SDS, 0.5% deoxycholate, and 1 mM EDTA in double distilled water) supplemented with cOmplete protease inhibitor cocktail (5892791001; Roche), PMSF (CAS 329-98-6; Sigma-Aldrich), and phosphatase inhibitor cocktails 1 and 2 (P5726; Sigma-Aldrich).
2) Triton-lysis buffer (150 mM NaCl, 10% glycerol, 1% Triton X-100, and 10 mM Tris, pH 7.4, in double distilled water) supplemented with cOmplete protease inhibitor cocktail, PMSF, and phosphatase inhibitor cocktails 1 and 2.

All the samples were lysed in RIPA buffer (1), except the samples from E13.5 embryos used to detect P-Ulk ser 555/Ulk tot, P-ACC/ACC tot, for which the lysis was performed with lysis buffer (2). Immunoblots for P-AMPK, P-Ulk, and LC3 I-II at E13.5 and P-Ulk and P-AMPK at E17.5 were loading-normalized by re-probing the original membranes with the respective loading control (tubulin $\beta III$ or $\beta$-actin) and total protein antibodies (total AMPK, Ulk, and tubulin $\beta III$ for LC3). The immunoblots were scanned and analyzed using ImageJ program. The samples for assessment of P-ACC/ACC at E13.5 were split and loaded on two pairs of identical gels. The intensities of the phosphorylated and the total forms detected on the first set of gels were then normalized on the tubulin $\beta III$ or $\beta$-actin as detected on the corresponding replica gel set, and then the values of the phosphorylated form/tubulin $\beta III$ or phosphorylated form/$\beta$-actin ratios were further adjusted relative to the ratios of total/tubulin$\beta III$ or total/$\beta$-actin.

### Immunostaining

After fixation with 4% formaldehyde. DRGs placed in the collagen matrix were gently washed with PBS. Blocking was done with 3% BSA (160069; MP Biomedicals, LLC) and 0.1% Triton X-100 (CAS900-93-1; Sigma-Aldrich) in PBS for 1 h. The blocking solution was used for incubation with primary and secondary antibodies. PBS washes were made between each step.

### Whole-mount embryo limbs

Limbs of E13.5 and E14.5 embryos were stained with anti-tubulin $\beta III$ antibody and anti-neurofilament antibody 2H3 following the iDisco methodology (Renier et al, 2014). For total axonal coverage and branching quantification, the NeuroMath software was used (Rishal et al, 2013). The numbers of samples processed in each experiment are specified in the corresponding figure legends.

### Gene expression microarray

DRGs from E13.5 *Lkb1* KO and WT embryos were harvested and RNA extracted as described in Rio et al (2010). For each sample, we pooled DRGs from two embryos. Overall, we analyzed the data from 10 microarrays, profiling five *Lkb1* KO and five WT samples. cDNA library construction and microarray hybridization were performed at the microarray unit of the Weizmann Institute of Science. Data were analyzed using MATLAB, false discovery rate 0.01.

Data are available at https://www.ncbi.nlm.nih.gov/geo/query/acc.cgi?acc=GSE146756.

### Immunohistochemistry of mouse embryos sections

E15.5–E17.5 embryos were fixed for 24 h in 4% formaldehyde and stained as described in Maor-nof et al (2016) with anti-Islet and anti-cleaved caspase-3 antibody. Numbers of embryos and sections used in individual experiments are specified in the figure legends.

The number of Iset1-positive neurons per DRG was calculated by computational approach as described in detail in Maor-nof et al (2016). The number of cleaved caspase-3–positive neurons was manually counted in ImageJ.

### Adult skin innervation

The hind limb glabrous skin of 2-mo-old mice was stained as described in Marvaldi et al (2015) (Zylka et al, 2005). For the nerve quantification, the fiber number per 150 $\mu m$ in the selected epidermis area was counted as described in Marvaldi et al (2015). The numbers of mice, limbs, and sections analyzed are specified in the figure legends.

### ATP measurement

Whole explants or axonal samples derived from *Lkb1* KO, *Efhd1* KO, and WT inserts were collected in 100 mM Tri and 4 mM EDTA, pH 7.75, and incubated for 2 min at 90°C. ATP concentration was measured using ATP Bioluminescence Assay Kit CLS II (11699695001; Roche). Whole protein level in the same samples were quantified (BCA protein assay kit; Pierce). The ATP measurements were then normalized to the protein concentration.

## Antibodies

Anti-tubulin βIII antibody (clone Tuj1; R&D Systems), at 1:20,000 for Western blot, 1:1,000 for immunoistochemistry (IHC) staining. P-AMPK thr172: phospho-AMPK-α (Thr172) (40H9) Cell Signaling, at 1:1,000. AMPK: AMPK-α (23A3) Cell Signaling, at 1:1,000. P-ACC ser 79: phospho-ACC (Ser79) Cell Signaling, at 1:1,000. ACC: anti-acetyl coenzyme A carboxylase antibody [EP687Y], AB-ab45174 Abcam, at 1:2,000. P-ULK ser555: phospho-ULK1 (Ser555) (D1H4) Cell Signaling, at 1:1,000. Caspase-3-cleaved: cleaved caspase-3 (Asp175) Cell Signaling, at 1:200. Islet antibody: Developmental Studies Hybridoma bank 39.3F7, at 1:200. Tom20: Santa Cruz Tom20 (FL-145): sc 11415, at 1:1,000 for immunofluorescence (STORM analysis) and TOM20 (D8T4N) Cell Signaling at 1:1,000 for immunoblot. β-Actin: β-Actin (13E5) Cell Signaling, at 1:20,000. Efhd1 antibody was a kind gift from the laboratory of Yasuhiro Tomooka (Tokyo University of Science), at 1:1,000. Anti-neurofilament antibody 2H3, Developmental Studies Hybridoma bank, at 1:200.

Antimouse and antirabbit antibodies conjugated with Alexa 549, Alexa 488, or Alexa 647 fluorophores were used at 1:200 (Jackson ImmunoResearch Laboratories). Secondary antibodies for Western blotting: goat antimouse IgG-HRP (JIR 155-035) and goat antirabbit IgG-HRP (JIR 111-035) from Jackson, both at 1:5,000.

## Stochastic optical reconstruction microscopy (STORM) imaging

DRGs from E13.5 embryos were plated on cell culture dish as described above.

Three-dimensional super-resolution images were recorded using a Vutara SR200 STORM microscope (Bruker) based on single-molecule localization biplane technology with 60× Olympus water-immersion objective (1.2 NA). Mitochondria (anti-TOM20 staining) labeled with AlexaFluor 647 were imaged using 640-nm excitation laser and 405-nm activation laser in an imaging buffer composed of 5 mM cysteamine, oxygen scavengers (7 μM glucose oxidase and 56 nM catalase) in 50 mM Tris with 10 mM NaCl, and 10% glucose at pH 8.0. Images were recorded using Evolve 512 EMCCD camera (Photometrics) with gain set at 50, frame rate at 50 Hz, and maximal power of 640 and 405 nm lasers set at 6 and 0.05 kW/cm$^2$, respectively. The total number of frames acquired was typically 15,000. Data analysis was performed using Vutara SRX software, localized particles were subjected to threshold value (set to 5) that is defined in Vutara SRX as standard deviations above the frame background value, which is determined based on the mean value of the border pixels in each frame. Mitochondrial length was manually analyzed using the SRX Vutara Software.

## Mitochondrial DNA quantification

DRGs were cultured in six-well chambers as described above. On the second day, the cultures were treated with FUDR (5-fluoro-2′-deoxyuridine, CAS 50-91-9, 100 nM; Merck) for 8 h. On the third day, FUDR was removed. On the fifth day, DNA was extracted with the MasterPure DNA purification kit (Cat. no. MCD85201; Epicentre), and quantitative real-time PCR was performed using SYBR Green (Cat. no. 4385612; Applied Biosystems). Expression levels were determined using the comparative cycle threshold ($2^{-\Delta\Delta Ct}$) method. 18S ribosomal RNA served as housekeeping genes.

Primers used were as follows:

**ND1**: FWD: 5′-TGCACCTACCCTATCACTCA-3′
REV: 5′-GCTCATCCTGATCATAGAATGG-3′
**ND1 and ND4**: FWD: 5′-CACTAATGCTACTACCACTAACCTGACTATC-3′
REV: 5′-TGTCATAGAAGTGTTAGGCTGGTTAA AC-3′
**Cytb1**: FWD: 5′-ACG TCC TTC CAT GAG GAC AA-3′
REV: 5′-GAG GTG AAC GAT TGC TAG GG-3′
**COX1**: FWD: 5′-GCCTTTGCTTCAAAACGAGA-3′
REV: 5′-GGTTGGTTCCTCGAATGTGT-3′
**MT-ATP8**: FWD: 5′-GCCACAACTAGATACATCAACATGA-3′
REV: 5′-GGTTGTTAGTGATTTTGGTGAAGGT-3′
**18S**: FWD-5′-AAACGGCTACCACATCCAAG-3′
REV-5′-CCTCCAATGGATCCTCGTTA-3′

## Mitochondrial respiration studies (Seahorse)

For the mitochondrial respiration studies, DRGs were dissected as described above, washed in HBSS, and dissociated by incubation in trypsin–EDTA solution B (03-052-1B; Biological Sciences) for 5 min. After trypsinization, dissociated DRGs were suspended in L15 plus serum and plated at a concentration of 30 × 10$^4$ on PDL/laminin (described above)–coated XF 96 plates (102601-100; Agilent) with neurobasal (described before). On the second day, FUDR (100 nm) was added to the media for 4 h to eliminate dividing cells, thus obtaining a purer neuronal culture. On day 4, mitochondrial respiration was examined using the Seahorse XF96 analyzer (Agilent) and the XF Cell Mito Stress Test Kit according to the manufacturer's instructions and as described in Karkucinska-Wieckowska et al (2015), Ruggiero et al (2017), Styr et al (2019). The analysis was performed in two parallel experiments as described in Styr et al., 2019. Briefly, in the first, respiration was measured under basal conditions and in response to the ATP synthases inhibitor oligomycin (1 μM), whereas in the second, respiration was measured under basal condition and in the presence of the electron transport chain accelerator ionophore FCCP (trifluorocarbonylcyanide phenylhydrazone, 4 μM). In both experiments, respiration was stopped by addition of the electron transport chain inhibitors rotenone and antimycin A (both at 0.5 μM). The neurons were then fixed with 4% formaldehyde and stained with tubulin βIII antibody. The plates were imaged, and the number of neurons was recalculated. Respiration values were normalized to the number of neurons. For each condition (oligomycin and rotenone/antimycin, and FCCP and rotenone/antimycin), we preformed four independent experiments; at least nine wells were used for each genotype, in each experiment. The baseline graph was generated by analyzing the third value of the baseline, in all the experiments. The ATP production graph was generated by subtracting the first value after oligomycin treatment from the third value of the baseline, in all the experiments. The SRC graph was generated by subtracting the third baseline value from the first value after FCCP injection, in all the experiments. The proton leak graph was generated by subtracting the first value after the injection of rotenone and antimycin from the third value after oligomycin injection, in all the experiments.

## Mitochondrial motility

E13.5-dissociated DRGs (described above) from *Lkb1* and *Efhd1* KOs and their respective control neurons were plated on PDL/laminin–coated microfluidic chambers. After 72 h, the mitochondria were labeled by 5 nM TMRE (tetramethylrhodamine ethyl ester; Thermo Fisher Scientific) in the *Lkb1* KO experiments and Mito-Tracker (M7514; Thermo Fisher Scientific) in the *Efhd1* KO experiments and imaged for 6 h. Motility analyses were performed by ImageJ as described in Ionescu et al (2016).

## Statistical analysis

Statistical analysis was performed using GraphPad Prism 7.0 software, Mean ± SEM is presented. The number of experiments (N) is indicated the figure legends. Normality of the data was defined using the Shapiro–Wilk normality test. For non-normally distributed data, Wilcoxon signed rank test and Mann–Whitney test were performed. For normally distributed homoscedastic data. two-tailed Student's *t* test was used and for non-homoscedastic *t* test with Welch's correction. For the Seahorse analysis, two-way ANOVA test and Sidak's multi-comparison test were performed.

# Supplementary Information

# Acknowledgements

We thank the Yaron lab members for advice and criticism, Eran Perlson and Tal Pery Gradus for the mitochondrial motility analysis, Ruggiero Antonella for consultation and advise on the Seahorse experiments and analysis, Vladimir Kiss for the help with the confocal microscope, Ron Rotkopf for excellent statistical assistance, and Andrew Kovalenko for critically reading the manuscript. This work was supported by funding to A Yaron from The Legacy Heritage Biomedical Science Partnership of the Israel Science Foundation (1004/09), National Science Foundation (NSF-IOS/BOI grant [1556968]), and The Nella and Leon Benoziyo Center for Neurological Diseases; Mr and Mrs James Kelly, Helene T Fox, Daniel C Andreae, the Samuel Aba, and Sisel Klurman Foundation in honor of Prof. Joel Sussman; the Advantage Trust in honor of Prof. Joel Sussman; and the Estate of Ethel Lena Levy. A Yaron is an incumbent of the Jack and Simon Djanogly Professorial Chair in Biochemistry.

## Author Contributions

V Ulisse: conceptualization, data curation, investigation, methodology, and writing—original draft, review, and editing.
S Dey: data curation and formal analysis.
DE Rothbard: data curation and investigation.
E Zeevi: data curation and investigation.
I Gokhman: data curation and investigation.
T Dadosh: data curation and investigation.
A Minis: conceptualization, data curation, and formal analysis.
A Yaron: conceptualization, resources, supervision, funding acquisition, project administration, and writing—original draft, review, and editing.

## Conflict of Interest Statement

The authors declare that they have no conflict of interest.

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
