## [Reviewer comments · Life Science Alliance]

Life Science Alliance

Regulation of axonal morphogenesis by the mitochondrial protein Efhd1

Valeria Ulisse, Swagata Dey, Deborah Rothbard, Einav Zeevi, Irena Gokhman, Tali Dadosh, Adi Minis, and Avraham Yaron

DOI: <https://doi.org/10.26508/lsa.202000753>

Corresponding author(s): Avraham Yaron, The Weizmann Institute of Science and Adi Minis, Rockefeller University

Review Timeline:	Submission Date:	2020-04-23
	Editorial Decision:	2020-04-24
	Revision Received:	2020-05-04
	Accepted:	2020-05-05

Scientific Editor: Andrea Leibfried

Transaction Report:

Please note that the manuscript was previously reviewed at another journal and the reports were taken into account in the decision-making process at Life Science Alliance.

Referee #1 Review

In this study, Ulisse and colleagues investigate the roles of Lkb1 and Efhd1 in the growth of dorsal root ganglia axons using in vitro and in vivo techniques in sensory neuron-specific knock-outs of Lkb1 and constitutive knock-outs of Efhd1 obtained via CRISPR-Cas9. They concluded that Efhd1 regulates axonal maturation and neuronal death probably downstream the LKB1/AMPK pathway. Authors also suggest that axonal growth and maintenance depend on the regulation by this pathway on the production of axonal ATP and activation of mitophagy. In general, I found the reported phenotypes very interesting, but the study remains very descriptive and several conclusions are at best correlative. The main conclusion that Efhd1 is regulated by Lkb1 is not supported by the data. Moreover, the rationale underlying the experiments is not always clear, and the experiments performed regarding the LKB1/AMPK pathway, mitochondrial and metabolic measurements and analysis of autophagy are not of adequate depth. I am confident these data could be improved but they will require an intensive amount of work; it is also possible that they could revert the conclusions stated here. Overall, although reporting a very interesting phenotype, the study fails to convincingly provide a mechanistic explanation for their findings.

Major comments:

- 1) There is no data in this paper to support the conclusion that Lkb1 regulates directly Efhd1. The only link is the downregulation of Efhd1 in Lkb1 deficient DRGs. Furthermore, there is no information about the other genes downregulated or upregulated. These data should be included in the manuscript.
- 2) The description of the Lkb1 KO phenotype remains open-ended.
- 3) The Efhd1 KO is a full body knock-out. What is the general phenotype of these mice?
- 4) I am very concerned about the statistical analysis of data: the low number of samples analysed in combination with the very high variation and low p values weakens my confidence in the data presented. For example, in Fig.5, I count 20 measurements with n=3. If authors counted 20 growth cones in a total of 3 different experiments, I really find difficult to trust the data, and the authors should be concerned as well. Another example is the counts of cleaved caspase positive cells. The authors counted 3 mice and with the low numbers detected, it is very likely that the obtained p value does not mean much (i.e. high false positive risk). It is likely that problems of this type underlie the discrepancy also between results in the two mouse models. The number of animals analysed and experiments performed should be at least doubled.
- 5) Western blots quantifications trigger doubts. It seems that the authors have normalized pAMPK to AMPK levels (and similar for other p-proteins). Since the two signals are coming from two different gels, they cannot be compared to each other. Normalization should include a housekeeping gene that is detected on the same membrane.
- 6) The only data presented here concern the % of stationary mitochondria. What about anterograde or retrograde moving mitochondria? Furthermore, there could be an impact on mitochondrial velocity. Authors should show corresponding kymographs.
- 7) Mitophagy experiments: the conclusion is only based on increased levels of p-Ulk1 (which are questionable, see point 5).
- 8) How are ATP measurements normalized? To the protein concentration (considering that less axonal growth characterizes the mutants)? Which are the absolute values? Again, n is too low.
- 9) The methods section should be carefully improved: please specify details about mouse models used (genetic backgrounds, diet, housing conditions according to the ARRIVE guidelines). Furthermore, details on the number of experiments, statistical analysis and quantification should be included.

Other comments:

- 1) The paper contains several typos, and language inconsistencies. It should be carefully checked and revised.
- 2) No data is presented to show that Lkb1 is deleted from DRG in the mouse model.
- 3) "Severely stunted axonal growth (36%) (Figure 1H) and even more pronounced decrease of the total number of axonal branches (48 %) (Figure 1I) were observed in the LKB1 KO limbs relative to the WT." The image is representative for branching but not for length. KO axons are not 50% shorter than WT.
- 4) "This visualization revealed mild axonal fragmentation at E13.5 (Supplementary Figure 1D, F)

and extensive axonal degeneration at E15.5 (Supplementary Figure 1E,G)."

The data presented seem to point out to nerve degeneration but they must be substantiated using cross sections of single nerves stained in different ways and with different markers.

5) "We could not detect any significant difference between LKB1 KO and control littermate neuronal numbers in this assay (Supplementary Figure 2A)". LKB1-deficiency increases proliferation in cancer and in neuronal precursors. Is it possible that DRG precursors overproliferate and then degenerate? This might explain the discrepancy observed in these data.

6) "LKB1 KO" and "LKB1-null mice". Inconsistent nomenclature.

7) "As judged by the extent of their phosphorylation (Supplementary Figure 2B-D)." From the images showed, it seems P-ACC and P-AMPK are active also in the soma. Better images should be presented. I encourage the authors to further prove their findings by immunofluorescence of P-AMPK and P-ACC in soma and axon.

8) "However, at E14.5 (Figure 3D-G), we did detect a significant 24% reduction in the total neurite length (Figure 3H) as well as a 28% decrease in the total number of branches (Figure 3I)."

As for the LKB1 characterization, the image is representative for branching but not for length. KO axons are not shorter than WT ones.

9) "No differences were noted between the numbers of neurons and the apoptotic rates in Efdh1 WT and KO DRGs at E15.5 (Supplementary Figure 4J, K)." To check apoptosis authors can perform a TUNEL assay. There are also Casp3-independent apoptosis mechanisms.

10) "The mitochondria in KO axons appeared significantly shortened compared to the WT organelles (0.77 μm vs 1.13 μm , respectively; Figure 5G-I)." The images do not mirror the graph. Moreover, if the quantification is correct, the difference in length is negligible. I would rather suggest that in the KO there are less mitochondria. Also here I am curious to know what the single measurements are referred to: is the number of experiments or the number of growth cones analysed? If the second case is true, the number should be drastically increased.

11) "The decreased ATP level observed in Efdh1 KO axons prompted us to examine the activation status of AMPK in the knockout cells." The rationale is wrong. AMPK is activated by increased AMP/ADP:ATP ratio. The absolute ATP value is not an indicator of AMPK activation. Fig.2B "Analysis of Efdh1 expression". Substitute mitocalcin with Efdh1. It's confusing.

12) Fig.4I :Graphs show mean {plus minus} SEM (t-test*p= 0.0134, n=3) (Scale Bar 100 μm).

What is n=3? The number of animals? How many pictures were analysed per animal?

13) Supp. Fig.1B: How many cells were analysed? Or is the average of different experiments?

14) Supp. Fig.1C: Brn3a, not Bren3a.

Referee #2 Review

The manuscript entitled " The mitochondrial protein Efhd1 is regulated by Liver Kinase B1 and is required for neuronal development" contains a number of potentially interesting observations. However, these observations are too preliminary and more work is needed to substantiate the conclusions.

Major:

1. Total ATP levels were decreased in the axonal fraction of LKB1 KO culture (Fig 1K) but this is proportional to decreased axonal length (Fig 1C) in that culture. This allows to suggest that there is less total ATP because there is less axons and not because the individual axons have less ATP. Same concerns Mitocalcin/Efhd1 KO axons. Note also that new ATP sensors ATEAM and Perceval would have been more appropriate for such type of experiments.
2. The authors did not detect any impact of LKB1 deletion on mitochondrial motility. However, they measured only the percentage of stationary mitochondria that is not a very sensitive parameter. Mitochondrial velocity or distances travelled would have been more informative.
3. Fig 1C shows that the axons in LKB1 KO are 50% shorter than in controls. Fig 2C shows that protein levels of Mitocalcin/Efhd1 are 60-70% lower in LKB1 KO than in control. Fig 3C shows that axonal length only 20% shorter in Mitocalcin/Efhd1 KO. This suggests that downregulation of Mitocalcin is only a minor contributor in the suppression of axonal growth in LKB1 KO. The authors identify also number of upregulated proteins and it can't be excluded that these upregulated proteins play an important role in suppressing axonal growth.
4. The authors suggest that the mitophagy might be activated in Efhd1 KO DRGs but do not provide experimental evidence (LC3, mito Keima).
5. The authors suggest that loss of Efhd1 inhibits mitochondrial function but do not provide experimental evidence to support that suggestion. ATP and mitochondrial length measurements are not sufficient to conclude that there will be less mitochondria or that they are not fully functional.
6. It remains unclear whether it is the downregulated Efhd1 that affects axonal growth in LKB1 KO (Fig 7A). Authors should perform rescue experiments to show that overexpression of Efhd1 in LKB1 KO will restore the axonal growth.

Minor:

1. There are no elongated mitochondria in growth cones. All mitochondria depicted in Fig5 C-I are round shape.
2. How exactly the axonal lengths were measured (Fig1C, H) is not well explained.
3. Blots presented in Figure 6 AB and Suppl Fig 2B do not have the controls (tubulin).
4. Experimental procedures are not sufficiently detailed.
5. For many experiments the number of replicates was 3 that is rather low.
6. The authors use only t - test throughout the paper. Did they check normality and heteroscedasticity of their data?

Referee #3 Review

In this study, Ulisse and colleagues uncover a new mitochondria-related mechanism for sensory neurons axonal outgrowth. Using *in vitro* and *in vivo* assays the authors show that knocking down the well characterized kinase LKB1 impaired axonal outgrowth of sensory neurons. Transcriptomic analysis of DRG neurons knock out for LKB1 reveals that a mitochondrial calcium binding protein called Efdh1/Mitocalcin is down regulated suggesting a mechanistic link between LKB1 and Efdh1. By taking advantage of a compartmentalised culture system the authors further show that this down regulation is axon specific similar to the ATP reduction they observed in LKB1 KO DRGs. Knocking down Efdh1 in mouse partially recapitulates the phenotype of the LKB1 KO which leads the authors to conclude that LKB1 regulates the development of sensory neurons via Efdh1. This study provides an interesting link between energy homeostasis and PNS axonal outgrowth.

The paper presents a well-designed and very focused study, at times almost minimalist, about neurodevelopment and especially the mechanisms that regulate axonal growth. The discovery of new pathways that help understanding how axons grow to reach their target has the potential to define new therapeutic targets to interfere with neurodevelopmental disorder and neurodegenerations. It is therefore highly significant. Furthermore, we only start to understand the crucial role played by mitochondria in neuronal development and this study brings yet another piece to this increasingly important field.

Among the most intriguing and interesting results presented in this paper are those suggesting that the mechanism that regulates axonal growth in DRG neurons via the LKB1/Efdh1 axis takes place locally in the axons. This implies a local regulation of mitochondrial function to support axonal growth during development which is a very innovative concept. This study also reveals a specific role for LKB1 in sensory neurons that has not been yet described and differs from the one observed in CNS neurons (polarization, mitochondrial transport). However, the data in their present form doesn't fully support the authors' conclusions. The manuscript is well written but lacks some depth in several occasions. In general, I support a straightforward style but this should not be to the detriment of important scientific data, references or explanations.

Major Concerns:

-The relationship between LKB1 and Efdh1 is intriguing but requires to be investigated in more details. Based on their transcriptomic analysis, the authors propose that LKB1 regulates Efdh1 transcriptionally. However, the authors didn't provide strong evidence for this transcriptional relationship. Therefore, the authors should complete their transcriptional analysis and test, for example, the level of Efdh1 in a context of LKB1 over-expression. A definitive evidence would be provided by a CHIP experiment to interrogate whether LKB1 interacts with the genomic regulatory elements of Efdh1. The potential transcriptional regulation of Efdh1 by LKB1 doesn't preclude that LKB1 could phosphorylate Efdh1 to modulate its functions. LKB1 main function being to phosphorylate its substrates I think it is worth investigating. Finally, the phenotypical similitude between neurons deleted for LKB1 and Efdh1 is also not sufficient to claim that they coordinate axonal development. One way to reinforce this point would be to over-express Efdh1 in LKB1 KO neurons and assess the level of rescue. Furthermore, if they act in the same pathway, the phenotype of a double KO shouldn't be more dramatic than any of the single KO.

- The authors claim that the mitochondrial transport is not affected by the deletion of LKB1 (Supp. Fig.2). However, they have not investigated the transport of mitochondria (or they are not showing all the data) in a comprehensive way. Mitochondrial transport needs to be evaluated entirely and all the transport parameters (moving frequency, speed, net travel...etc.) should be added. Due to the known role of LKB1 in the mitochondrial transport (Courchet et al. 2013) of cortical neurons and because the role of LKB1 in sensory neurons presented in this study seems to be different, mitochondrial transport in LKB1 KO and Efdh1 KO needs to be more carefully evaluated. The technique used by the author to assess mitochondrial transport limits greatly the scope of their analysis: (1) Mitochondria are labelled using TMRE which labels most if not all the mitochondria of the neurons in culture making difficult to assess which part of the axons is imaged. Sparse transfection of fluorescent protein targeted to the mitochondria (MitoDsRed or

mitoGFP) is preferable especially in DRG culture that can be easily transfected. (2) The method used to assess stationary and motile mitochondria is elegant but is far to offer a complete picture of all the transport parameters that one would get using a kymograph analysis or other object tracking software. Along the same line, this can be assess in vivo by looking a possible decrease of mitochondria at the axon's terminal. By looking at the representative images of the figure 5, it seems like Efdh1 KO growth cones have not only shorter mitochondria but they are less numerous which could indicate a transport impairment. Overall, a better characterization of mitochondrial transport in these neurons is needed to rule out that it is not affected by LKB1 and/or Efdh1 depletion.

-Although this study wants to elucidate the role of Efdh1 in sensory neuron axonal development, how Efdh1 affects mitochondrial physiology during this process is not investigated. For example is mitochondrial calcium homeostasis and bioenergetic impaired in Efdh1 KO neurons?

-The full microarray data presented in figure 2 should be included in the paper as supplemental material.

Minor concerns:

-Typos:

Page 6, paragraph 3: "Therefore we preformed transcriptome profiling (...)"

Page 9, beginning of paragraph 3 first sentence: "(...) although some to the same extend" .
"Some" is most likely not the intended word.

-The authors briefly address the phenotype of the Efdh1 mouse in the discussion but it should be explained in more details in the results.

-When the author described their experiments in adult mice (page 7, paragraphe 3), an age should be provided.

-Figure 2B: The western blot legend indicates Mitocalcin. Nomenclature should be consistent throughout the manuscript and this figure is the only time Mitocalcin is used instead of Efdh1.

2nd Round of Reviewer Reports from Other Journal

Referee #1 Review

Report for Author:

In this revised version, the authors have included novel experiments to address some of the previous criticisms, and focused the manuscript on the role of Efhd1 in sensory axon. The manuscript is improved and the data presented are solid. Overall, this study demonstrates the role of Efhd1 in axonal morphogenesis. However, this work does not really unravel the molecular function of Efhd1, and does not explain why loss of this mitochondrial protein leads to defects in axonal development. Given the new data showing that loss of Efhd1 affects mitochondrial ATP production and respiration, it is possible that this phenotype is the reflection of a generic mitochondrial dysfunction. The authors discuss this very honestly in the manuscript. The link between Lkb1 and Efhd1 expression remains suggestive, but not further explored.

I still have some comments:

1) The authors have clarified in the rebuttal letter the n for each experiment. However, how this is described in the Figure legend is still not clear. The authors should specify in each case the n, and

then explain how each independent biological value has been obtained. For instance, in the Legend to Figure 2K, the authors write: "5 WT and 7 Efhd1 KO embryos were analyzed and the number of CC3 positive was quantified in 60 sections/embryo (K)." Is n 5 and 7? In Figure 3 I, the N in the KO is less than 25.

2) The authors find reduced mRNA levels of Efhd1 in Lkb1 KO embryos. Interestingly, they found reduced levels of Efhd1 protein specifically in axons and not in the soma of Lkb1 sensory neurons. What about mRNA levels in soma and axons? Why is Efhd1 expression specifically affected in axons? This is potentially very interesting, but not explored at all. Are other mitochondrial proteins expressed at normal levels in Lkb1 sensory axons?

Referee #3 Review

Report for Author:

Authors have adequately responded to all of my comments and included a number of new experiments that support their conclusions. The revised manuscript is also restructured and focussed to EFHD1 that has considerably increased its clarity.

My only remaining (minor) concern is related to statistics. In the revised version the authors have checked whether their data did not follow the normal distribution. However, the authors should also test whether their data are homoscedastic and use t test with Welch's correction if the SD-s are not equal. This test is available in Prism 7 as well.

We thank the reviewers for their constructive comments. We agree that all of them raised many valid points, although we think some of the criticism (on the number of samples or Western Blot analysis) stemmed from the way we presented the data and information being missed by the reviewers. We have now finally completed a series of new experiments and analyses. Based on these new results, we also changed the focus of the paper to the identification of the mitochondrial protein EFHD1 as a regulator of axonal morphogenesis. Furthermore, the manuscript was extensively revised for better structure and clarity.

Below please find point by point response to all of the reviewers' comments.

Referee #1

In this study, Ulisse and colleagues investigate the roles of Lkb1 and Efhd1 in the growth of dorsal root ganglia axons using in vitro and in vivo techniques in sensory neuron-specific knock-outs of Lkb1 and constitutive knock-outs of Efhd1 obtained via CRISPR-Cas9. They concluded that Efhd1 regulates axonal maturation and neuronal death probably downstream the LKB1/AMPK pathway. Authors also suggest that axonal growth and maintenance depend on the regulation by this pathway on the production of axonal ATP and activation of mitophagy. In general, I found the reported phenotypes very interesting, but the study remains very descriptive and several conclusions are at best correlative. The main conclusion that Efhd1 is regulated by Lkb1 is not supported by the data. Moreover, the rationale underlying the experiments is not always clear, and the experiments performed regarding the LKB1/AMPK pathway, mitochondrial and metabolic measurements and analysis of autophagy are not of adequate depth. I am confident these data could be improved but they will require an intensive amount of work; it is also possible that they could revert the conclusions stated here. Overall, although reporting a very interesting phenotype, the study fails to convincingly provide a mechanistic explanation for their findings.

We thank the reviewer, who found our work very interesting. The reviewer raises two points: the conclusion that Efhd1 is regulated by LKB1 is not supported by the data and that the experiments lack depth. To address the first point, we took a pharmacological approach and now show **new data in Figure 1 and Supplementary figure 2** that acute inhibition of AMPK by Comp-C *in vitro* induces a strong reduction in the level of Efhd1, but not of the mitochondrial protein Tom20. Broadly, we toned down the conclusion that Efhd1 is directly regulated by LKB1, as we believe this is not a critical part of the paper any more, whose main focus now is the identification of Efhd1 as a novel regulator of axonal morphogenesis. For the second point, we have performed new experiments, as outlined below, that we believe enhance the initial conclusion of the paper and provide additional depth.

Major comments:

1) There is no data in this paper to support the conclusion that Lkb1 regulates directly Efhd1. The only link is the downregulation of Efhd1 in Lkb1 deficient DRGs. Furthermore, there is no information about the other genes downregulated or upregulated. These data should be included in the manuscript.

Please see above our response to this point. We now provide the names of all the deregulated genes on the Volcano plot. Once accepted will provide an Excel file with the complete microarray data and deposit all the raw data.

2) The description of the Lkb1 KO phenotype remains open-ended.

We agree, but as noted above, characterization of the Lkb1 KO merely opened the door to the analysis of the Efhd1, which is the main focus of this paper.

3) The Efhd1 KO is a full body knock-out. What is the general phenotype of these mice?

The Efh1 KO are viable and fertile, of regular size and appearance. Observation of the mice in their home cages didn't reveal any aberrant behavior.

4) I am very concerned about the statistical analysis of data: the low number of samples analyzed in combination with the very high variation and low p values weakens my confidence in the data presented. For example, in Fig.5, I count 20 measurements with n=3. If authors counted 20 growth cones in a total of 3 different experiments, I really find difficult to trust the data, and the authors should be concerned as well. Another example is the counts of cleaved caspase positive cells. The authors counted 3 mice and with the low numbers detected, it is very likely that the obtained p value does not mean much (i.e. high false positive risk). It is likely that problems of this type underlie the discrepancy also between results in the two mouse models. The number of animals analyzed and experiments performed should be at least doubled.

We have now specified the exact number of samples that we analyzed in each experiment in the figure legends; also see below for some examples. In the vast majority of the experiments, we analyzed multiple embryos and DRGs generated from 3 independent litters and therefore the *n* was mistakenly understated in the original manuscript.

In Fig 5 what are the exact measurements: how many embryos? how many GCs? How many mitochondria?

Overall, we measured about 150 mitochondria of each genotype in 25 GCs randomly selected from different DRGs of different embryos (KO and WT littermate), in 3 independent experiments. I would like to point that these numbers are in line with other studies in the field for example see Seok-Kyu et.al *Plos Biology* (2016) and Lewis et.al *Nat.Comm* (2018).

In Fig-2 (axonal length *in vitro*), we measured the axonal length of 8 WT and 8 Efh1 KO DRGs by four measurements of each DRG in 3 independent experiments (overall 24 DRGs).

In Fig-2 (limbs axonal morphology *in vivo*), seven WT and eight Efh1 KO embryos were analyzed, data for each embryo represents the average measurements of both limbs.

In Fig-2 (islet1 and CC3), we analyzed five WT and seven Efh1 KO embryos, cells were counted in 60 sections/embryo.

We believe that the “*discrepancy between the two mouse models*” is driven by the biology.

The LKB1 pathway regulates multiple energetic processes, Efh1 being an effector in just one of them. Therefore, it would be naïve to expect ablation of Efh1 to mimic the strong phenotypes we observed in the LKB1 KO. Indeed, our analysis suggests that in the Efh1 KO the LKB1-AMPK pathway is intact, as we detected the activation of AMPK and its downstream effector ULK1 in the Efh1 KO neurons, presumably due to the mitochondria defects.

5) Western blots quantifications trigger doubts. It seems that the authors have normalized pAMPK to AMPK levels (and similar for other p-proteins). Since the two signals are coming from two different gels, they cannot be compared to each other. Normalization should include a housekeeping gene that is detected on the same membrane.

The reviewer is of course correct regarding the normalization and the quantification methods and this is exactly how it was performed, as now outlined in the methods section. We have not included the anti-tubulin data in the original Figs for tidiness' sake, but noted that in the methods section. In the **new Supplementary Figure 5**, we show extended Figures with the tubulin loading controls for all the Western blots.

6) The only data presented here concern the % of stationary mitochondria. What about anterograde or retrograde moving mitochondria? Furthermore, there could be an impact on mitochondrial velocity. Authors should show corresponding kymographs.

We now provide a more complete analysis of the mitochondria motility (stationary, anterograde and retrograde) including kymographs for both LKB1 and Efh1 KO. We should note that the most dramatic effect of LKB1 KO on mitochondrial motility in cortical neurons observed by Courchet et.al was on the % of *stationary mitochondria*, which clearly doesn't change.

7) *Mitophagy experiments: the conclusion is only based on increased levels of p-Ulk1 (which are questionable, see point 5).*

We now provide **new data in Figure 5** demonstrating increase in autophagic flux in Ehfd1 KO neurons, further supporting the idea that mitophagy is enhanced in the knockout neurons.

8) *How are ATP measurements normalized? To the protein concentration (considering that less axonal growth characterizes the mutants)? Which are the absolute values? Again, n is too low.*

The ATP measurements were normalized to the protein concentration.

I would like to elaborate on the culture systems that we used in this study. In principle, we used two types of cultures: two-dimensional cultures, in which the neurons are plated on PDL/Laminin, and three-dimensional cultures, in which the neurons are embedded in collagen. We have detected reduced axonal growth only in the three-dimensional cultures, due to the fact that it is more restrictive (**see new data in Supplementary Figure 4 and Figure 2**).

Therefore, the two-dimensional cultures were used for the biochemical experiments.

The N for these important experiments is 12 independent replicates, which is not low by any standard, and is now clearly stated in the legends.

9) *The methods section should be carefully improved: please specify details about mouse models used (genetic backgrounds, diet, housing conditions according to the ARRIVE guidelines). Furthermore, details on the number of experiments, statistical analysis and quantification should be included.*

We have now extended the methods and the figure legends sections providing additional details on the mice, number of experiments, statistics analysis and quantifications.

Other comments:

1) *The paper contains several typos, and language inconsistencies. It should be carefully checked and revised.*

The paper was edited.

2) *No data is presented to show that Lkb1 is deleted from DRG in the mouse model.*

We now provide **new data in Supplementary Figure 1** that clearly demonstrate the ablation of LKB1 in DRGs by the Wnt-1 Cre.

3) *"Severely stunted axonal growth (36%) (Figure 1H) and even more pronounced decrease of the total number of axonal branches (48 %) (Figure 1I) were observed in the LKB1 KO limbs relative to the WT." The image is representative for branching but not for length. KO axons are not 50% shorter than WT.*

We agree with the reviewer that the definitions of the phenotypes were not accurate. We now use two parameters: overall axonal coverage (total axonal length that cover the limb's surface) and total number of branches. We believe these definitions faithfully describe the phenotypes.

4) *"This visualization revealed mild axonal fragmentation at E13.5 (Supplementary Figure 1D, F) and extensive axonal degeneration at E15.5 (Supplementary Figure 1E,G)."*

The data presented seem to point out to nerve degeneration but they must be substantiated using cross sections of single nerves stained in different ways and with different markers.

We decided to omit these data from the new version of the manuscript, as it is more focused on Ehfd1

5) *"We could not detect any significant difference between LKB1 KO and control littermate neuronal numbers in this assay (Supplementary Figure 2A)". LKB1-deficiency increases proliferation in cancer and in neuronal precursors. Is it possible that DRG precursors over proliferate and then degenerate? This might explain the discrepancy observed in these data.*

We do not consider the possibility that DRG precursors over-proliferate and then degenerate as a likely scenario, as we detect normal cell number (Islet-1 positive) in the KO at E15.5. Moreover, our *in vitro* DRGs cultures do not contain neuronal precursors and yet we clearly detect axonal phenotypes.

6) "LKB1 KO" and "LKB1-null mice". Inconsistent nomenclature.

The nomenclature is now consistent.

7) "As judged by the extent of their phosphorylation (Supplementary Figure 2B-D)." From the images showed, it seems P-ACC and P-AMPK are active also in the soma. Better images should be presented. I encourage the authors to further prove their findings by immunofluorescence of P-AMPK and P-ACC in soma and axon.

We now provide the images with the loading control, quantification was done as requested in Major Comments 5 above. Indeed, ACC and AMPK are also active in the soma. However, our quantifications did not pick any significant decrease there. We have not been able to detect, in a reliable manner, P-AMPK and P-ACC in soma and axons by immunofluorescence.

8) "However, at E14.5 (Figure 3D-G), we did detect a significant 24% reduction in the total neurite length (Figure 3H) as well as a 28% decrease in the total number of branches (Figure 3I)."

As for the LKB1 characterization, the image is representative for branching but not for length. KO axons are not shorter than WT ones.

We agree with this comment, please see the response to point 3.

9) "No differences were noted between the numbers of neurons and the apoptotic rates in Efh1 WT and KO DRGs at E15.5 (Supplementary Figure 4J, K)." To check apoptosis authors can perform a Tunel assay. There are also Casp3-independent apoptosis mechanisms.

We agree with the reviewer and therefore performed two complementary types of analysis: the anti-cleaved caspase-3 staining, which is a traditional assay for classical apoptosis, and the count of neuronal number (Islet1 positive). With the latter approach, we would detect a reduction in neuronal cell number regardless of the mechanism of cell death.

10) "The mitochondria in KO axons appeared significantly shortened compared to the WT organelles (0.77 μm vs 1.13 μm , respectively; Figure 5G-I)." The images do not mirror the graph. Moreover, if the quantification is correct, the difference in length is negligible. I would rather suggest that in the KO there are less mitochondria. Also here I am curious to know what the single measurements are referred to: is the number of experiments or the number of growth cones analysed? If the second case is true, the number should be drastically increased.

The image represents the differences in mitochondria length, but we agree with the reviewer that it is misleading regarding the mitochondria number.

The points represent independent GCs. Overall, 150 mitochondria of each genotype were quantified in ~25 GCs.

I would like to point that these numbers are in line with other studies in the field for example see Seok-Kyu et al *Plos Biology* (2016) and Lewis et al *Nat. Comm* (2018).

We now address the issue of mitochondria number by two additional approaches:

1. Direct counting of the mitochondria in GCs (new data in Figure 3)
2. Global analysis using Q-PCR of the select mitochondrial genome targets, which is considered a gold standard for mitochondria quantification (Ruggiero et al. 2017; Maryanovich et al. 2015; Costanzini et al 2019; Giordano et al 2014) (new data in Figure 3). By both approaches, we did not detect a statistically significant reduction in mitochondria number.

Lastly, we performed mitochondria motility analysis and did not observe any aberrances of the mitochondrial transport in the Efh1 KO neurons (new data in Figure 4).

Importantly, we did discover significant mitochondria dysfunction using Seahorse XF analyzer (new data in Figure 4).

Overall, these new results support the idea that ablation of Efh1 causes mitochondrial malfunction but does not affect their number or motility.

11) "The decreased ATP level observed in Efh1 KO axons prompted us to examine the activation status of AMPK in the knockout cells." The rationale is wrong. AMPK is activated by increased AMP/ADP:ATP ratio. The absolute ATP value is not an indicator of AMPK activation. Fig.2B "Analysis of Efh1 expression". Substitute mitocalcin with Efh1. It's confusing. We agree with the reviewer and corrected the text accordingly.

12) Fig.4I :Graphs show mean {plus minus} SEM (t-test*p= 0.0134, n=3) (Scale Bar100µm). What is n=3? The number of animals? How many pictures were analyzed per animal? We have analyzed three animals of each genotype, 25 sections per animal.

13) Supp. Fig.1B: How many cells were analysed? Or is the average of different experiments? The results are the average of four experiments, around 30 cells per experiment were analyzed. Note that we co-plated the cells to avoid any effect of cell density.

14) Supp. Fig.1C: Brn3a, not Bren3a.
Fixed.

Referee #2:

The manuscript entitled " The mitochondrial protein Efh1 is regulated by Liver Kinase B1 and is required for neuronal development" contains a number of potentially interesting observations. However, these observations are too preliminary and more work is needed to substantiate the conclusions.

We thank the reviewer, who found our work" potentially interesting". We hope the reviewer will agree that the new data substantiate the conclusions.

Major:

1. *Total ATP levels were decreased in the axonal fraction of LKB1 KO culture (Fig 1K) but this is proportional to decreased axonal length (Fig 1C) in that culture. This allows to suggest that there is less total ATP because there is less axons and not because the individual axons have less ATP. Same concerns Mitocalcin/Efh1 KO axons. Note also that new ATP sensors ATEAM and Perceval would have been more appropriate for such type of experiments.*

The ATP measurements were normalized to protein concentration. See also our detailed response to Referee #1 regarding axonal growth under these culture conditions. Therefore, we believe the measurements are consistent with a reduction in ATP levels.

2. *The authors did not detect any impact of LKB1 deletion on mitochondrial motility. However, they measured only the percentage of stationary mitochondria that is not a very sensitive parameter. Mitochondrial velocity or distances travelled would have been more informative.*

We now provide a more comprehensive analysis of the mitochondrial motility (stationary, anterograde and retrograde), including kymographs for both the LKB1 KO and the Efh1 KO. We should note that the most dramatic effect of LKB1 KO on mitochondrial motility in cortical neurons observed by Courchet et.al was on the % of *stationary mitochondria*, which clearly doesn't change in our system. We do not think that changes in mitochondrial velocity or distances travelled might explain the severe reduction in ATP levels observed in the KOs. In contrast our finding that the mitochondrial activity is reduced provides very good explanation.

3. Fig 1C shows that the axons in LKB1 KO are 50% shorter than in controls. Fig 2C shows that protein levels of Mitocalcin/Efh1 are 60-70% lower in LKB1 KO than in control. Fig 3C shows that axonal length only 20% shorter in Mitocalcin/Efh1 KO. This suggests that downregulation of Mitocalcin is only a minor contributor in the suppression of axonal growth in LKB1 KO. The

authors identify also number of upregulated proteins and it can't be excluded that these upregulated proteins play an important role in suppressing axonal growth.

We agree with the reviewer's comment and would like to add that the LKB1 pathway regulates multiple energetic processes. Therefore, it will be naïve to expect ablation of Efh1 to mimic the strong phenotypes we observed in the LKB1 KO. Moreover, our analysis suggests that this comparison is complicated, as in the Efh1 KO the LKB1-AMPK pathway is intact, and we detected the activation of AMPK and its downstream effector ULK1 in the Efh1 KO neurons, presumably due to the mitochondria defects. In the revised paper, we have toned down the statements on the importance of Efh1 as an LKB1 effector and focused on the identification of Efh1 as a regulator of axonal morphogenesis.

4. The authors suggest that the mitophagy might be activated in Efh1 KO DRGs but do not provide experimental evidence (LC3, mito Keima).

We now provide new data in Figure 5, by quantifying LC3 levels, that support an increase in autophagic flux in the Efh1 KO DRGs.

5. The authors suggest that loss of Efh1 inhibits mitochondrial function but do not provide experimental evidence to support that suggestion. ATP and mitochondrial length measurements are not sufficient to conclude that there will be less mitochondria or that they are not fully functional.

We now provide new data in Figure 4 on mitochondrial functionality obtained with Seahorse XF analyzer that clearly show mitochondria dysfunction in the Efh1 KO. Additional new data in Figure 4 address the issue of mitochondrial number and motility.

6. It remains unclear whether it is the downregulated Efh1 that affects axonal growth in LKB1 KO (Fig 7A). Authors should perform rescue experiments to show that overexpression of Efh1 in LKB1 KO will restore the axonal growth.

Rescue experiments will involve the generation of a Tg that expresses Efh1 and analysis of the animals on the background of the Wnt1-Cre/LKB1cKO. Since the main focus of the paper is now the identification of Efh1 as a regulator of axonal morphogenesis, we have not pursued this line of experiments.

Minor:

1. There are no elongated mitochondria in growth cones. All mitochondria depicted in Fig5 C-I are round shape.

In general, the mitochondria in growth cones do not assume the highly elongated shape commonly observed in fibroblasts.

2. How exactly the axonal lengths were measured (Fig1C, H) is not well explained.

The axonal morphology was analyzed by Neuromath (Rishal et al. 2013).

3. Blots presented in Figure 6 AB and Suppl Fig 2B do not have the controls (tubulin).

We now provide the tubulin loading controls for all the blots see supplementary Figure 5.

4. Experimental procedures are not sufficiently detailed.

We have extended the experimental procedures section.

5. For many experiments the number of replicates was 3 that is rather low.

For most experiments N is much higher than 3, see above the response to Reviewer #1.

6. The authors use only t - test throughout the paper. Did they check normality and heteroscedasticity of their data?

Based on the reviewer comments we re-looked into the statistical analysis.

We performed the analysis using Graph-Pad Prism 7.0 software. In the legend of each figure the number of experiments is now indicated. Normality of the sample was defined with Shapiro-Wilk normality test. For not normally distributed data Wilcoxon Signed rank test and Mann Whitney test were performed. For normally distributed data the two-tailed Student's t-test was used. For the Sea Horse analysis Two Way ANOVA test and Sidak's Multi Comparison Test was performed. This description of the statistical analysis is now part of the Experimental procedures section.

We should note that this re-analysis didn't change any of the claims that we made based on the previous analysis.

Referee #3:

In this study, Ulisse and colleagues uncovers a new mitochondria-related mechanism for sensory neurons axonal outgrowth. Using in vitro and in vivo assays the authors show that knocking down the well characterized kinase LKB1 impaired axonal outgrowth of sensory neurons. Transcriptomic analysis of DRG neurons knock out for LKB1 reveals that a mitochondrial calcium binding protein called Efdh1/Mitocalcin is down regulated suggesting a mechanistic link between LKB1 and Efdh1. By taking advantage of a compartmentalised culture system the authors further show that this down regulation is axon specific similar to the ATP reduction they observed in LKB1 KO DRGs. Knocking down Efdh1 in mouse partially recapitulates the phenotype of the LKB1 KO which leads the authors to conclude that LKB1 regulates the development of sensory neurons via E1hd1. This study provide an interesting link between energy homeostasis and PNS axonal outgrowth.

The paper presents a well-designed and very focused study, at time almost minimalist, about neurodevelopment and especially the mechanisms that regulate axonal growth. The discovery of new pathways that help understanding how axons grow to reach their target has the potential to define new therapeutic targets to interfere with neurodevelopmental disorder and neurodegenerations. It is therefore highly significant. Furthermore, we only start to understand the crucial role played by mitochondria in neuronal development and this study bring yet another piece to this increasingly important field.

Among the most intriguing and interesting results presented in this paper are those suggesting that the mechanism that regulates axonal growth in DRG neurons via the LKB1/Efdh1 axis takes place locally in the axons. This imply a local regulation of mitochondrial function to support axonal growth during development which is a very innovative concept. This study also reveals a specific role for LKB1 in sensory neurons that has not been yet described and differs from the one observed in CNS neurons (polarization, mitochondrial transport). However, the data in their present form doesn't fully support the authors' conclusions. The manuscript is well written but lack some depth in several occasions. In general, I support a straightforward style but this should not be to the detriment of important scientific data, references or explanations.

We thank the reviewer for the positive and encouraging comments.

Major Concerns:

-The relationship between LKB1 and Efdh1 is intriguing but requires to be investigated in more details. Based on their transcriptomic analysis, the authors propose that LKB1 regulate Efdh1 transcriptionally. However, the authors didn't provide strong evidence for this transcriptional relationship. Therefore, the authors should complete their transcriptional analysis and test, for example, the level of Efdh1 in a context of LKB1 over-expression. A definitive evidence would be

provided by a CHIP experiment to interrogate whether LKB1 interacts with the genomic regulatory elements of Efdh1. The potential transcriptional regulation of Efdh1 by LKB1 doesn't preclude that LKB1 could phosphorylates Efdh1 to modulate its functions. LKB1 main function being to phosphorylate its substrates I think it is worth investigating. Finally, the phenotypical similitude between neurons deleted for LKB1 and Efdh1 is also not sufficient to claim a that they coordinate axonal development . One way to reinforce this point would be to over-express Efdh1 in LKB1 KO neurons and assess the level of rescue. Furthermore, if they act in the same pathway, the phenotype of a double KO shouldn't be more dramatic than any of the single KO.

We agree with the reviewer and decided to change the focus of the paper to Efdh1 as a new regulator of axonal morphogenesis rather than its possible role as a downstream effector of LKB1. Therefore, we toned down the claim that Efdh1 is directly regulated by LKB1 and generated additional data, as outlined above, on the function of Efdh1.

- The authors claim that the mitochondrial transport is not affected by the deletion of LKB1 (Supp. Fig.2). However, they have not investigated the transport of mitochondria (or they are not showing all the data) in a comprehensive way. Mitochondrial transport needs to be evaluated entirely and all the transport parameters (moving frequency, speed, net travel...etc.) should be added. Due to the known role of LKB1 in the mitochondrial transport (Courchet et al. 2013) of cortical neurons and because the role of LKB1 in sensory neurons presented in this study seems to be different, mitochondrial transport in LKB1 KO and Efdh1 KO needs to be more carefully evaluated. The technique used by the author to assess mitochondrial transport limit greatly the scope of their analysis: (1) Mitochondria are labelled using TMRE which labels most if not all the mitochondria of the neurons in culture making difficult to assess which part of the axons is imaged. Sparse transfection of fluorescent protein targeted to the mitochondria (MitoDsRed or mitoGFP) is preferable especially in DRG culture that can be easily transfected. (2) The method used to assess stationary and motile mitochondria is elegant but is far to offer a complete picture of all the transport parameters that one would get using a kymograph analysis or other object tracking software. Along the same line, this can be assess in vivo by looking a possible decrease of mitochondria at the axon's terminal. By looking at the representative images of the figure 5, it seems like Efdh1 KO growth cones have not only shorter mitochondria but they are less numerous which could indicate a transport impairment. Overall, a better characterization of mitochondrial transport in these neurons is needed to rule out that it is not affected by LKB1 and/or Efdh1 depletion.

It is known that the role of LKB1 is not conserved in all types of neurons. For example, LKB1 is required for axonal formation in cortical neurons but not in many other types of neurons including sensory (Lilley et.al 2013 and our study). Therefore, we are not completely surprised that the role of LKB1 in mitochondrial transport is not conserved. We should note that the most dramatic effect of LKB1 KO on mitochondrial motility in cortical neurons observed by Courchet et.al was on the % of *stationary mitochondria*, which clearly doesn't change. However, we agree with the reviewer that additional characterization of mitochondrial transport in the LKB1 and the Efdh1 KOs is important and we now provide these as **new data in Figure 4**. For our studies on mitochondrial motility in the Efdh1 KO we used MitoTracker, which is not sensitive to mitochondrial activity. We cannot rule out changes in other parameters but we think that they are less likely to be in the basis of the phenotypes as the decrease in mitochondrial activity.

-Although this study wants to elucidate the role of Efdh1 in sensory neuron axonal development, how Efdh1 affects mitochondrial physiology during this process is not investigated. For example is mitochondrial calcium homeostasis and bioenergetic impaired in Efdh1 KO neurons?

We now provide **new data in Figure 4** of mitochondrial function by Seahorse analysis in the Efdh1 KO, which clearly demonstrate mitochondrial dysfunction.

-The full microarray data presented in figure 2 should be included in the paper as supplemental

material.

We now indicate the names of the deregulated genes on the Volcano plot. Once accepted will provide an Excel file with the complete microarray data and deposit all the raw data.

Minor concerns:

-Typos:

Page 6, paragraph 3: "Therefore we performed transcriptome profiling (...)"

Page 9, beginning of paragraph 3 first sentence: "(...) although some to the same extend" .

"Some" is most likely not the intended word.

These typos were fixed

-The authors briefly address the phenotype of the Efdh1 mouse in the discussion but it should be explained in more details in the results.

We now address the phenotype of the Efdh1 mouse in the results part as well.

-When the author described their experiments in adult mice (page 7, paragraphe 3), an age should be provided.

We now provide the age of these animals

-Figure 2B: The western blot legend indicates Mitocalcin. Nomenclature should be consistent throughout the manuscript and this figure is the only time Mitocalcin is used instead of Efdh1.

These typos were fixed

April 24, 2020

RE: Life Science Alliance Manuscript #LSA-2020-00753-T

Prof. Avraham Yaron
The Weizmann Institute of Science
Biological Chemistry department
234 Herzl
Rehovot 76100
Israel

Dear Dr. Yaron,

Thank you for transferring your revised manuscript entitled "Regulation of axonal morphogenesis by the mitochondrial protein Efhd1". Your manuscript was reviewed by the same reviewers twice before, and the editors transferred those reports to us with your permission.

The reviewers appreciate your data and the revision performed, but would have expected more mechanistic insight into the potential functional relationship between LKB1 and Efhd1. Lack thereof does not preclude publication here, and we would thus be happy to publish your paper in Life Science Alliance pending final minor revisions:

- please address the remaining concern of rev#1, point 1 and rev#3, point on statistics
- all corresponding authors should link their ORCID iDs to their profiles in the submission system, please (instructions on how to do so have been sent by email)
- please list 10 authors et al in the reference list
- please provide source data for Fig. 5H
- please add callouts in the ms text to Fig 3B
- please deposit the microarray data and add accession code information to the M&M section

A. FINAL FILES:

B. MANUSCRIPT ORGANIZATION AND FORMATTING:

Thank you for your attention to these final processing requirements.

Sincerely,

Andrea Leibfried, PhD
Executive Editor
Life Science Alliance
Meyerhofstr. 1
69117 Heidelberg, Germany
t +49 6221 8891 502

e.a.leibfried@life-science-alliance.org
www.life-science-alliance.org

- please address the remaining concern of rev#1, point 1 and rev#3, point on statistics
Rev#1, point Point-1:

1) The authors have clarified in the rebuttal letter the n for each experiment. However, how this is described in the Figure legend is still not clear. The authors should specify in each case the n, and then explain how each independent biological value has been obtained. For instance, in the Legend to Figure 2K, the authors write: "5 WT and 7 Efh1 KO embryos were analyzed and the number of CC3 positive was quantified in 60 sections/embryo (K)." Is n 5 and 7? In Figure 3 I, the N in the KO is less than 25.

We now specify in every figure legend the N that was used for the graphs and the statistical analysis. The N for KO in figure 3I is 20 and not 25.

Rev#3

My only remaining (minor) concern is related to statistics. In the revised version the authors have checked whether their data did not follow the normal distribution. However, the authors should also test whether their data are homoscedastic and use t test with Welch's correction if the SD-s are not equal. This test is available in Prism 7 as well.

We have analyzed the data for homoscedasticity and found three places in which t-test with Welch's correction was required, outlined in the Figure legends. Importantly, this methodology didn't change the outcome the statistical analysis.

- all corresponding authors should link their ORCID iDs to their profiles in the submission system, please (instructions on how to do so have been sent by email)

We provide the ORCID iDs for the corresponding authors

- please list 10 authors et al in the reference list

The reference list is corrected.

- please provide source data for Fig. 5H

We now provide a source data for all the W.Bs figures.

- please add callouts in the ms text to Fig 3B

We inserted a callout for Figure-3B in the text.

- please deposit the microarray data and add accession code information to the M&M section

The Microarray data was deposited and the accession number is provided in the M&M, under the microarray section.

May 5, 2020

RE: Life Science Alliance Manuscript #LSA-2020-00753-TR

Prof. Avraham Yaron
The Weizmann Institute of Science
Biological Chemistry department
234 Herzl
Rehovot 76100
Israel

Dear Dr. Yaron,

Thank you for submitting your Research Article entitled "Regulation of axonal morphogenesis by the mitochondrial protein Efh1". I appreciate the introduced changes and it is a pleasure to let you know that your manuscript is now accepted for publication in Life Science Alliance. Congratulations on this interesting work.

DISTRIBUTION OF MATERIALS:

Again, congratulations on a very nice paper. I hope you found the review process to be constructive and are pleased with how the manuscript was handled editorially. We look forward to future exciting submissions from your lab.

Sincerely,

Andrea Leibfried, PhD
Executive Editor
Life Science Alliance
Meyerohofstr. 1
69117 Heidelberg, Germany
t +49 6221 8891 502
e a.leibfried@life-science-alliance.org
www.life-science-alliance.org